# Accounting for Missing Covariates in Heterogeneous Treatment Estimation

**Khurram Yamin**                                                                *kyamin@cmu.edu*
*Department of Machine Learning*
*Carnegie Mellon University*

**Vibbhu Sharma**                                                      *vibhhus@andrew.cmu.edu*
*Department of Computer Science*
*Cornell University*

**Edward Kennedy**                                                          *edward@stat.cmu.edu*
*Department of Statistics*
*Carnegie Mellon University*

**Bryan Wilder**                                                                *bwilder@cmu.edu*
*Department of Machine Learning*
*Carnegie Mellon University*

## Abstract

Many applications of causal inference require using treatment effects estimated on a study population to then make decisions for a separate target population that lacks treatment and outcome data. We consider the challenging setting where there are important covariates that are observed in the target population but are missing from the original study. Our goal is to estimate the tightest possible bounds on heterogeneous treatment effects conditioned on such newly observed covariates. We introduce a novel partial identification strategy based on ideas from ecological inference; the main idea is that estimates of conditional treatment effects for the full covariate set must marginalize correctly when restricted to only the covariates observed in both populations. Furthermore, we introduce a bias-corrected estimator for these bounds and prove that it enjoys fast convergence rates and statistical guarantees (e.g., asymptotic normality). Experimental results on both real and synthetic data demonstrate that our framework can produce bounds that are much tighter than would otherwise be possible.

## 1 Introduction

Many applications of causal inference require using treatment effects estimated on a study population to then make decisions for a separate target population that lacks treatment and outcome data. As a motivating example, consider a health system that wishes to deploy a new intervention in its population while making use of existing study data that was used to estimate heterogeneous treatment effects. The health system will almost certainly have access to features that were not measured in the original study, due to differences in institutional settings. For example, if the initial study was an RCT (Randomized Control Trial), it may have failed to measure practically important covariates such as social determinants of health (Kahan et al., 2014; Huang et al., 2024). Since the intervention has not previously been used by the health system, no outcome data linked to these new covariates is available. However, treatment decisions would ideally reflect whether the intervention is likely to be beneficial to a patient conditional on *all* information available, not just covariates that happened to be in the original study. This paper studies the question: how precisely can we identify treatment effects conditional on such new covariates? If precise estimates are available, the decision maker can proceed confidently with deployment. Conversely, if considerable uncertainty remains about an

important subgroup, a decision maker may exercise more caution or invest more resources in monitoring or additional data collection.

Formally, we aim to derive bounds on conditional average treatment effects (CATEs) when novel covariates are observed in the target population. We refer to the CATE conditional on both the common and new covariates as the *fully conditional* CATE and the CATE conditional on only the common covariates (which is what can be estimated from the original study) as the *restricted* CATE. Intuitively, what makes informative bounds possible is that the fully conditional CATE must be consistent with the restricted CATE when marginalized to only the common covariates. This idea is reminiscent of the ecological inference literature, which focuses on inferring a joint distribution from its marginals. Ecological inference has long been used in the quantitative social sciences, e.g. for election analysis (Glynn & Wakefield, 2010; King et al., 2004). However, almost no previous work uses ideas from ecological inference in causal settings. We provide a partial identification strategy new to causal inference by connecting ideas from ecological inference to causality. The resulting bounds on treatment effects use the joint distribution of the common and new covariates to link the fully conditional and restricted CATEs.

We make the following contributions. First, we formally provide provable bounds on conditional treatment effects by leveraging ideas from ecological inference. These bounds contain novel nuisance functions that must be estimated. Our second contribution is a bias-corrected estimator that exhibits favorable statistical properties such as allowing for the use of non-parametric and/or slow converging machine learning models to estimate these nuisance functions without sacrificing fast $O_{\mathbb{P}}(\frac{1}{\sqrt{n}})$ rates convergence. We also prove that our estimator is asymptotically normal, facilitating the construction of confidence intervals. Finally, we demonstrate these properties empirically through the use of simulation and application to data from a real RCT.

**Additional related work:** There is a great deal of work that focuses on combining experimental and observational data to estimate treatment effects. This paper does *not* focus on the distinction between experimental and observational data: our setup is agnostic as to whether the study population is experimental or observational as long as it satisfies standard identification assumptions. Our focus is on incorporating *covariates* that are newly observed in the target population and not present in the study. One major line of previous work attempts to use outcome data from both an RCT and observational study to jointly estimate treatment effects (on common covariates). (Chen et al., 2021; Hatt et al., 2022; Demirel et al., 2024; Guo et al., 2022; Schweisthal et al., 2024) Often, this involves fitting a model for the confounding bias present in the observational study (Kallus et al., 2018; Yang et al., 2020; Wu & Yang, 2022). By contrast, we estimate the CATE in the target population where no outcome data is available and new significant covariates are present that were not present in the study population (we do not only study common covariates between data sources). A second line of work focuses on the case where the observational study has no outcomes by correcting for shift in the covariate distribution (Lesko et al., 2017; Lee et al., 2022). These methods deal only with covariates found in common, and focus on average effects (as opposed to our focus on conditional effects).

Recently, a few papers have tried to quantify the uncertainty from covariates missing in the RCT. Colnet et al. (2024) focus on quantifying how an estimate of the ATE might be biased when a covariate is missing in one or both populations. In contrast, our focus is on bounding conditional treatment effects. They also require distributional assumptions (e.g., Gaussianity) on the missing covariate, as opposed to our nonparametric approach. Similarly, Nguyen et al. (2016) propose a sensitivity analysis framework for estimation of the ATE in a target population when a treatment effect mediator is unobserved in the target population (as opposed to our focus on covariates unobserved in the study population). Andrews & Oster (2019) propose a framework to assess the external validity of RCTs when a missing covariate induces selection bias into the trial.

Several works make strong assumptions that allow for point identification when covariates are missing. Pearl (2012) studies noisy measurements $W$ that causally depend on latent covariates $Z$, enabling point identification by assuming $W$ contains no additional necessary information beyond $Z$. In contrast, we consider novel covariates unique to the target domain that cannot be reconstructed from study data, thus precluding point identification and requiring bounds. Similarly, Bareinboim & Pearl (2014) achieves point identification by assuming access to a transportability graph that models causal links between all variables across domains

Table 1: Notation and Definitions (Not Exhaustive)

| Symbol | Definition |
| --- | --- |
| $V$ | Common covariates observed in both study and target populations. |
| $W$ | Additional (discrete) covariates observed only in the target population. |
| $T \in \{0,1\}$ | Treatment assignment indicator in the study population. |
| $Y$ | Observed outcome in the study population. |
| $Y^1, Y^0$ | Potential outcomes under treatment (1) and control (0). |
| $E \in \{0,1\}$ | Population indicator: $E=1$ for study, $E=0$ for target. |
| $Z$ | Observed data tuple $\{V, E, W(1-E), E(T,Y)\}$. |
| $X$ | Shorthand for the full covariate vector $(V, W)$. |
| $\mu_t(v)$ | $\mathbb{E}[Y \mid V=v, T=t, E=1]$, $t \in \{0,1\}$. |
| $\nu(v,w)$ | $\mathbb{P}\big(W=w \mid V=v, E=0\big)$. |
| $\tau_\ell(x), \tau_u(x)$ | Un-truncated lower/upper bound on CATE numerator: $\big(\mu_1 - \mu_0 \pm (b-a)\big)/\nu$. |
| $\gamma_\ell(v,w), \gamma_u(v,w)$ | Truncated lower/upper bounds on $\mathbb{E}[Y^1 - Y^0 \mid v, w, E=1]$ (Thm. 1). |
| $m(X;\beta)$ | model class for Parametric projection of the bounds. |
| $\beta$ | Parameters of the chosen model class for $m(\cdot; \beta)$. |
| $h(X)$ | Analyst-chosen weight function in the projection. |
| $g(X)$ | $\big(\partial m / \partial \beta\big) h(X)$ in the moment conditions. |
| $\pi_t(v)$ | Joint propensity $\mathbb{P}(T=t, E=1 \mid V=v)$, $t \in \{0,1\}$. |
| $\rho_0(v)$ | Selection probability $\mathbb{P}(E=0 \mid V=v)$. |
| $f(X)$ | Indicator $\mathbf{1}\{\tau_\ell(X) + b - a \geq 0\}$ (or analogously for upper). |
| $a, b$ | Known bounds on $Y$: $Y \in [a,b] \implies Y^1 - Y^0 \in [a-b, b-a]$. |
| $\delta$ | Sensitivity parameter: $\big|\mathbb{E}[Y^1 - Y^0 \mid V, W] - \mathbb{E}[Y^1 - Y^0 \mid V]\big| \leq \delta$. |

– an assumption we do not make. Berrevoets et al. (2023) enables point identification by making an overlap assumption where all covariates and sets of covariates have observed outcome values with non-zero probability. In our setting however, we have certain covariates in our target population that we never have outcome data for. To our knowledge, our paper is the first work to ever non-parametrically estimate bounds on the CATE in this setting, where the target population has no observed data and has covariates not observed in the study population, without making the kind of strong assumptions that allow point-identification while compromising real-world applicability.

## 2  Problem Setup

We examine the situation where we are attempting to transport estimates of heterogeneous treatment effects between two populations. In the first population, which we refer to as the *study* population, we observe covariates $V$, treatment assignments $T \in \{0,1\}$, and outcomes $Y$. Each individual has potential outcomes $Y^1$ and $Y^0$ that would be realized if they were (respectively, were not) treated. We assume that $Y^1, Y^0 \in [a,b]$ with probability 1 for some constants $a$ and $b$, i.e., the outcomes are bounded between known values. We observe $Y^T$ corresponding to the treatment assignment. In this population, we impose standard identifying assumptions (most prominently, no unobserved confounding) that allow estimation of the conditional average treatment effect (CATE) $\mathbb{E}[Y^1 - Y^0 | V]$ of the study population which we refer to as the *restricted* CATE. This study population could represent a randomized experiment or an unconfounded observational setting. This assumption is formalized as $T \perp\!\!\!\perp Y^t \mid V$ given the context of the study population.

In the second population, which we refer to as the *target* population, we do not observe the treatment or outcome variables. Instead, we observe just the covariates, which consist of both $V$ and a new set of covariates $W$ which were not observed in the study. We assume that $W$ consists only of discrete covariates (although $V$ may be either discrete or continuous). This holds naturally in many settings of practical interest (e.g., social determinants of health are very often discrete variables (Sarkar, 2014)). Furthermore, this assumption is not overly restrictive it can otherwise be ensured via discretization of continuous values (e.g., many clinical risk scales are already discretized into a fixed set of levels). (Ustun & Rudin, 2019) This assumption is technically

required so the probability of a specific realization of $W$ is well-defined. We use an indicator variable $E$ to indicate whether the subject is in the study population ($E = 1$) or the target population ($E = 0$). Our observed data consists of samples

$$Z = \left\{V, E, W(1-E), E(T, Y)\right\} = \begin{cases} V, W & : E = 0 \\ V, T, Y & : E = 1 \end{cases}$$

Our goal is to estimate $\mathbb{E}[Y^1 - Y^0 | V, W]$, or the CATE conditioned on both $V$ and $W$. We refer to this quantity as the *fully conditional* CATE. We are going to use the standard assumptions of consistency ($Y=Y^t$ whenever $T = t$), positivity ($\mathbb{P}(T = t | V = v, W = w) > 0$ for all combinations of $v, w$). We also assume that (1) $\mathbb{P}(W = w \mid V = v, E = 1) = \mathbb{P}(W = w \mid V = v, E = 0)$, and (2) $\mathbb{E}(Y^1 - Y^0 \mid V, W, E = 1) = \mathbb{E}(Y^1 - Y^0 \mid V, W, E = 0)$. These two assumptions could be restated as (1) the covariates missing in the study population have the same conditional distribution as those in the target population, and (2) the fully conditional CATE ($\mathbb{E}[Y^1 - Y^0 \mid V, W]$) is the same in the study and target population. Many notations and symbols are summarized in Table 1.

Furthermore, it is true that the treatment effects given $V$ are unconfounded based on data from the study population. However, unconfoundedness given $V$ simply means that $V$ gives sufficient information to understand treatment choices in the study population. It does not mean that $V$ contains all variables that impact treatment effects for specific individuals. As such, while we can successfully estimate the CATE conditional just on $V$, different subgroups *within* a stratum of $V$ may still have heterogeneous treatment responses. In our model, $W$ represents important additional covariates that create smaller subgroups within strata of $V$. We wish to bound the treatment effects for these smaller subgroups as they can better allow us to tailor treatments to individuals at a more personalized level.

## 3 Methodology

In order to estimate treatment effect bounds in the presence of covariates observed only in the target population, our approach proceeds in three key stages. First, we invoke an ecological–inference argument to derive sharp, nonparametric bounds on the fully conditional CATE $\mathbb{E}[Y^1 - Y^0 \mid V, W]$ by leveraging the fact that these effects must aggregate to the known restricted CATE given $V$ alone. Next, recognizing that these bounds depend on several unknown nuisance functions (e.g. outcome regressions, propensity scores, and the conditional distribution of $W$), we develop a bias-corrected estimator based on influence-function adjustments: by carefully correcting the usual plug-in estimator, we attain fast $O(n^{-1/2})$ convergence rates even when the nuisances are learned flexibly at slower, nonparametric rates. Finally, to further tailor our inferences to settings where one believes the additional covariates $W$ have quantifiable limited impact (by domain knowledge), we introduce a simple sensitivity-analysis model that restricts the deviation between the restricted CATE (using only $V$) and fully-conditional CATE (using $V, W$) by a user-specified parameter $\delta$, yielding strictly tighter bounds when plausible. These steps are summarized in Figure 1. We now describe each of these steps in turn.

### 3.1 Partial identification bounds

Our goal is to provide as much information as possible about the fully conditional CATE (i.e., conditional on both $V$ and $W$) when outcome data is linked only to $V$. Clearly, in this setting it is not possible to exactly identify the fully conditional CATE. However, we can use ideas from ecological inference to *partially* identify it. Specifically, if we average the fully conditional CATE over values of $W$, we must obtain the CATE conditioned only on $V$, which *is* identified. Formally, a long line of work in ecological inference (Jiang et al., 2019; Plescia & De Sio, 2017; Manski, 2016) uses marginal consistency conditions of the following form, for a single outcome $Y$:

$$\mathbb{E}(Y \mid v) = \sum_w \mathbb{E}(Y \mid v, w) p(w \mid v). \tag{1}$$

For the sake of brevity, we use capital letters (eg $V, W$) to represent random variables and lowercase versions (e.g. $v, w$) to represent specific values of these letters (such that $V = v, W = w$). If $p(w \mid v)$ is known or can

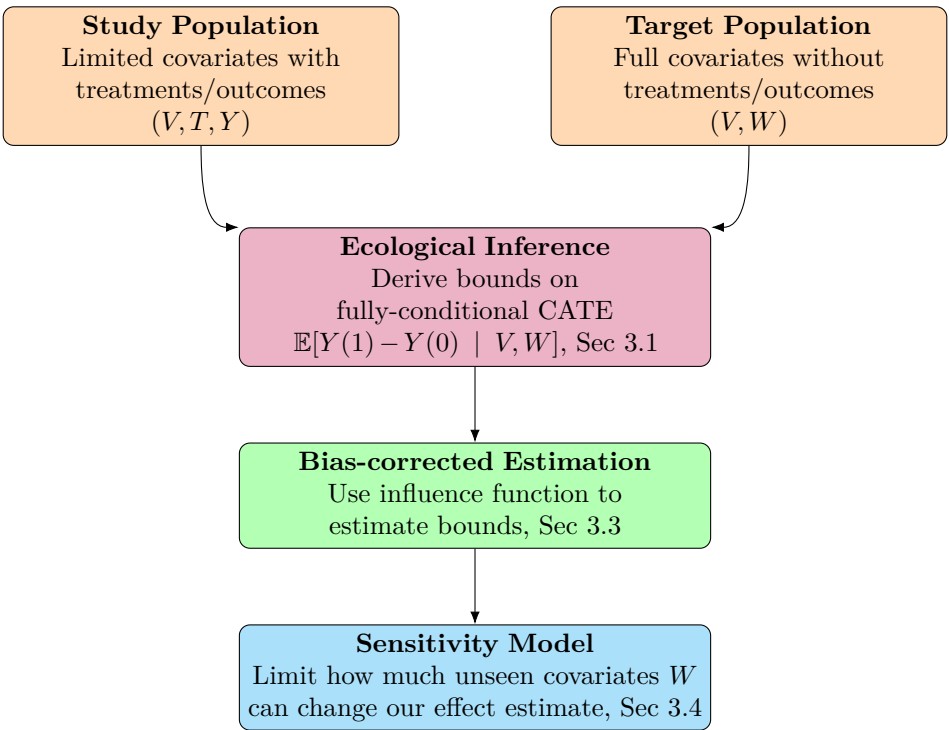

Figure 1: Roadmap for bounding and estimating the fully-conditional CATE

be estimated, we can then rearrange this expression to obtain bounds on $E(Y \mid v, w)$. We apply a similar strategy in the context of the CATE (using also the causal assumptions that link it to the observable data) and obtain the following provable bounds on the fully conditional CATE:

**Theorem 1.** *Assuming the conditions in Section 2 and that $Y$ is a real-valued outcome bounded in [a,b]*

$$\gamma_\ell(v, w) \leq \mathbb{E}(Y^1 - Y^0 \mid v, w, E = 1) \leq \gamma_u(v, w)$$

*such that*

$$\gamma_\ell(v, w) = \max\left\{\frac{\mu_1(v) - \mu_0(v) - (b - a)(1 - \nu)}{\nu}, a - b\right\}$$

$$\gamma_u(v, w) = \min\left\{\frac{\mu_1(v) - \mu_0(v) - (a - b)(1 - \nu)}{\nu}, b - a\right\}$$

*where $\mu_t(v) = \mathbb{E}(Y \mid V = v, T = t, E = 1), \nu(v, w) = \mathbb{P}(W = w \mid V = v, E = 0)$*

This bound utilizes the worst-case scenario that if $Y \in [a, b]$, then $Y^1 - Y^0 \in [a - b, b - a]$. Note that these bounds (our estimand of interest) are a function of $V$ and $W$. We introduce a framework that estimates the projection of these functions onto a parameterized model class $m$ chosen by the analyst. Different use cases may call for different parameterizations. E.g., an analyst may prefer a linear model for simplicity and ease of interpretation, or use a more intricate nonlinear model such as a neural network if they wish for greater flexibility to capture the underlying function. Regardless of the choice, we assume that the model is parameterized by some $\beta$ of fixed dimension with respect to $n$. Note that we do not assume that the model $m$ is correctly specified; i.e. we estimate the projection of $\gamma$ onto a model class instead of setting $\gamma$ equal to some model.

For simplicity of notation, we will use $X$ interchangeably with the joint tuple $(V, W)$. We attempt to find the best approximation of $\gamma_\ell(x)$ and $\gamma_u(x)$ in the form of a model $m$. We use $\gamma$ to represent the bound of interest, which could be either $\gamma_\ell(x)$ or $\gamma_u(x)$ as desired. We seek model parameters $\beta$ which minimize the

mean squared error between $m$ and $\gamma$:

$$\beta = \arg\min_{\theta \in \mathbb{R}^p} \mathbb{E}\left[h(X)\{\gamma(X) - m(X;\theta)\}^2\right]$$

This formulation is weighted by an analyst-chosen function $h$ which allows a degree of choice on what observations to place emphasis on (for example, we could choose to emphasize certain covariate regions of interest). Differentiating then gives us the moment condition $M(\beta)$

$$0 = \mathbb{E}\left[\frac{\partial m(X;\beta)}{\partial \beta} h(X)\{\gamma(X) - m(X;\beta)\}\right] = M(\beta)$$

Our goal becomes to find the $\hat{\beta}$ such that $M(\hat{\beta}) = 0$. To keep notation concise, define

$$\tau_\ell(x) = \frac{\mu_1(v) - \mu_0(v) - (b - a)(1 - \nu)}{\nu}$$

so that $\gamma_\ell(x) = \max\{\tau_\ell(x), a - b\}$, and similarly for $\tau_u(x)$. Letting $g(X) = \frac{\partial m(X;\beta)}{\partial \beta} h(X)$, this moment condition is equivalent to

$$M_\ell(\beta_\ell) = \mathbb{E}[g(X)\{(\tau_\ell(X) + b - a) * \mathbf{1}(\tau_\ell(X) + b - a \geq 0) + a - b - m(X;\beta)\}] = 0 \tag{2}$$

$$M_u(\beta_u) = \mathbb{E}[g(X)\{(\tau_u(X) + a - b) * \mathbf{1}(\tau_u(X) + a - b \leq 0) + b - a - m(X;\beta)\}] = 0 \tag{3}$$

### 3.2 Estimation

If $\tau$ were known, we could simply solve the resulting least-squares problem. However, $\tau$ in fact depends on a number of nuisance functions that are not known and must be estimated from the data: $\hat{\mu}_1(V), \hat{\mu}_0(V)$, and $\hat{\mathbb{P}}(W = w \mid V = v, E = 0)$. A naive plug-in strategy would be to estimate each of the nuisance functions and then plug the estimates into the moment condition.

However, the quality of the resulting solution will depend sharply on how well the nuisances are estimated. In general we will not obtain consistent estimates for even the projection onto the parametric model class unless the nuisances are estimated consistently. As the nuisances are unlikely to lie exactly in any specific parametric class, consistent estimation will require the use of nonparametric methods that converge only slowly (slower than $O(n^{-\frac{1}{2}})$). Conversely, if the true values of the nuisance functions were known, $\beta$ could be estimated at the (faster) root-$n$ parametric rate (Vaart, 2000). We draw on techniques from the semiparametric statistics/double ML literature to propose a Bias-Corrected estimator that attains the parametric rate for $\beta$ even when the nuisance functions are estimated at slower nonparametric rates.

### 3.3 Bias-Corrected Estimator

In this section, we will focus on the derivation of a Bias-Corrected estimator for $\gamma_\ell(x)$; $\gamma_u(x)$ follows a similar form, shown in the Appendix. Full proofs of all claims can be found in the Appendix. The starting point is to derive an *influence function* for our estimand of interest. Intuitively, influence functions approximate how errors in nuisance function estimation impact the quantity of interest, in this case $\gamma$. A common strategy in semiparametric statistics is to use the influence function for the target quantity to provide a first-order correction for the bias introduced by nuisance estimation. This dampens the sensitivity of the estimator to errors in the nuisances and will allow us to derive fast convergence rates for $\beta$ even when the nuisances converge more slowly.

Unfortunately, influence functions typically only exist for quantities that are pathwise differentiable. The expression for $\gamma_\ell$ contains a non-differentiable max, which shows up as an indicator function in the moment $M_\ell(\beta_\ell)$ (2). That is, the moment condition is discontinuous on the margin $\tau(x) + b - a = 0$ to $\tau_\ell(x) + b - a < 0$. We employ a margin condition strategy used by Kennedy et al. (2019), who analyzed a nondifferentiable instrumental variable model, and others (Kpotufe et al., 2022; Vigogna et al., 2022). Specifically, we can hope for fast estimation rates when the classification problem of estimating the indicator for a given $X$ is not

too hard, in the sense that not too much probability mass is concentrated near the boundary. Specifically, we assume that

$$\textbf{Assumption (Margin Condition): } \mathbb{P}(|\tau_\ell(x) + b - a| \leq \zeta) \leq C\zeta^\alpha \qquad (4)$$

for some constant C and some $\alpha \geq 0$. Similar assumptions are often imposed in the context of classification problems (Audibert & Tsybakov, 2007). We have $\alpha = 0$ with no further assumptions, and $\alpha = 1$ holds if $\tau(x)$ has a bounded density. A bounded density is a relatively weak assumption, so $\alpha \geq 1$ is likely to hold in many cases of interest. Under this margin condition, we employ a two-part strategy. First, we derive an influence function for the moment condition under the assumption that the true value of the indicator function is known, rendering the expression differentiable in the estimated nuisances. Second, our final estimator replaces the indicator function with a plug-in estimate; the margin condition entails that this step introduces relatively small bias compared to if the true indicator were known. Specifically, we prove that

$$\mathbb{E}[\tau_\ell \mathbf{1}(\hat{\tau}_\ell(X) + b - a) - \tau_\ell \mathbf{1}(\tau_\ell(X) + b - a)] \leq C||\hat{\tau}_\ell(X) - \tau_\ell(X)||_\infty^{1+\alpha}$$

When $\alpha \geq 1$, this term now depends only on squared errors in the estimation of $\tau$ and will be negligible asymptotically so long as $\hat{\tau}_\ell$ converges at a $o(n^{-\frac{1}{4}})$ rate. The proof for this is contained in the appendix. For simplicity of notation, we call the indicator $f(X)$ such that $f(X) = \mathbf{1}(\tau(x) + b - a \geq 0)$. Next, we turn to deriving the influence function for $\gamma$ under the assumption that the indicator $f$ is known (where our eventual estimator will use the analysis above to justify replacing $f$ with its plugin estimate). Given this strategy, we derive the influence function $\varphi(X, \beta, \eta)$ (full derivation in Appendix) to obtain:

**Lemma 1.** $\varphi(X, \beta, \eta)$ is given by (assuming conditions from Section 2):

$$\sum_w \begin{bmatrix} V \\ w \end{bmatrix} f(V, w) \{ \frac{ET}{\pi_1(V)}(Y - \mu_1(V)) - \frac{E(1-T)}{\pi_0(V)}(Y - \mu_0(V)) \} + \frac{(b-a)(1-E)}{\rho_0(V)}$$

$$\{ X f(X) - \sum_w (f * \nu)(V, w) \} - \frac{1-E}{\rho_0(V)} \{ X f(X) \tau_\ell(X) - \sum_w \begin{bmatrix} V \\ w \end{bmatrix} (f * \tau_\ell * \nu)(V, w) \}$$

$$+ X \{ \tau_\ell(X) f(X) + (b-a) f(X) + a - b - m(X, \beta) \}$$

where $\pi_t(V) = p(T = t, E = 1|V)$, $\rho_0(V) = p(E = 0|V)$

Note that $\pi$ and $\rho$ appear as new nuisance functions that help compensate for errors in the original plug-in estimate. From this influence function, we can construct our bias-corrected estimator. The key idea is to find $\hat{\beta}$ that solves an estimating equation implied by the influence function:

$$\hat{\beta} \text{ such that } P_n\{\varphi(X, \hat{\beta}, \hat{\eta})\} = 0$$

where $P_n$ is an empirical average. Formally, in order to construct the estimator, we employ a sample splitting procedure detailed in Algorithm 1. This follows the strategy, common in semiparametric inference Kennedy et al. (2023), of splitting the dataset into two independent halves. The first is used to estimate the nuisance functions (including the indicator $f$). Then, we fix the nuisances and construct $\varphi(X, \hat{\beta}, \hat{\eta})$ for the points in the second half, which are used to estimate $\beta$ via the above moment condition. The computational approach to solving the moment condition will depend on the model family chosen. For linear models $m(v, w, \beta) = \beta_1^T v + \beta_2^T w$, we give a closed-form solution in the appendix that can be computed as a standard OLS problem. For more general differentiable model classes, one strategy would be to minimize $P_n\{\varphi(X, \hat{\beta}, \hat{\eta})^2\}$ using gradient-based methods.

We now turn to analyzing the convergence properties of the bias-corrected estimator, with the goal of showing that $\hat{\beta} \to \beta$ at a fast rate even when the nuisances are estimated slowly. We start by examining the bias $R_n$ of our influence function. Let $\eta_0$ be the true values of the nuisance functions while $\hat{\eta}$ is our estimate. The bias $R_n$ quantifies the difference between the expected influence function at $\eta_0$ and $\hat{\eta}$ and plays a key role in controlling the convergence rate of $\hat{\beta}$. Formally, we decompose the bias as:

**Theorem 2.** Let $R_n = \mathbb{P}\{\varphi(X; \beta, \hat{\eta}) - \varphi(X; \beta, \eta_0)\}$. Assuming that all nuisance functions and their estimates are bounded below by a constant larger than 0 and that all probabilities are bounded above by 1,

$$R_n \lesssim \|\hat{\rho} - \rho\|_2 \|\hat{\nu} - \nu\|_2 + \|\hat{\nu} - \nu\|_2^2 + \|\hat{\pi}_1 - \pi_1\|_2 \|\hat{\mu}_1 - \mu_1\|_2$$

$$+ \|\hat{\pi}_0 - \pi_0\|_2 \|\hat{\mu}_0 - \mu_0\|_2 + \|\mu_1 - \hat{\mu}_1\|_2 \|\hat{\nu} - \nu\|_2 + \|\hat{\mu}_0 - \mu_0\|_2 \|\hat{\nu} - \nu\|_2$$

Roughly, our estimated $\hat{\beta}$ will converge quickly if $R_n = o_{\mathbb{P}}(\frac{1}{\sqrt{n}})$ (a statement formalized below). For this to occur, all of the products above must be $o_{\mathbb{P}}(\frac{1}{\sqrt{n}})$. It is a sufficient but not a necessary condition that $\hat{\rho}, \hat{\nu}, \hat{\pi}$, and $\hat{\mu}$ converge to their true functions at $o_{\mathbb{P}}(n^{-\frac{1}{4}})$ rates for $R_n$ to be $o_{\mathbb{P}}(\frac{1}{\sqrt{n}})$. Note that $n^{-\frac{1}{4}}$ is substantially slower than the parametric root-$n$ rate, and is satisfied by even many nonparametric methods. It becomes clear from the bias structure that $\hat{\nu} = \hat{P}(W = w | V = v, E = 0)$ would need to be correctly specified for $\hat{\beta}$ to be a consistent estimator of $\beta$. This is due to the nature of the ecological inference setting where $\hat{\nu}$ is the only thing linking the target population to the study. Thus, our estimator is not doubly robust in the sense that no other nuisance can compensate for errors in $\hat{\nu}$. However, the error is still second-order as it involves only squared errors for $\nu$. Additionally, there is a mixed-bias property with respect to all of the other nuisances, where each nuisance can converge at a slower rate individually if others converge faster (e.g., $\mu$ can converge more slowly if $\pi$ and $\nu$ converge faster). Therefore, our estimator exhibits robustness to misspecification in all nuisances except $\nu$. Putting these pieces together, we obtain a convergence guarantee for $\hat{\beta}$:

**Theorem 3.** *If $R_n = O_{\mathbb{P}}(\frac{1}{\sqrt{n}})$ and assuming the conditions in Section 2 and that the margin condition (4) holds for $\alpha \geq 1$:*

$$||\hat{\beta} - \beta||_2 = O_{\mathbb{P}}\left(\frac{1}{\sqrt{n}}\right) \qquad and \qquad \sqrt{n}(\hat{\beta} - \beta) \xrightarrow{D} N\left(0, M^{-1}VM^{-1}\right)$$

*where $M = \frac{\partial \mathbb{E}(\varphi(X; \beta, \eta_0))}{\partial \beta^T}$ and $V$ is the variance of $\varphi(X; \beta, \eta_0)$*

Theorem 3 follows from Theorem 2 combined with a standard analysis of $M$-estimators under misspecification (c.f. Theorem 5.2.1 of Vaart (2000) and Lemma 3 of Kennedy et al. (2023)). Asymptotic normality is valuable as it allows for the construction of confidence intervals, e.g. with the usual sandwich estimator (or potentially easier in practice, the bootstrap).

---

**Algorithm 1:** Algorithm for Constructing Bias-Corrected Estimator

---

1: Given input samples $\mathcal{D}$, split uniformly at random into $\mathcal{D}_1$ and $\mathcal{D}_2$
2: Use $\mathcal{D}_1$ to estimate nuisance functions: $\hat{\tau}_\ell(X), \hat{\mu}_1(X), \hat{\mu}_0(X), \hat{\pi}_1(X), \hat{\pi}_0(X), \hat{\rho}_0(X), \hat{\nu}(X)$
3: Use the estimated nuisances estimates to construct the estimated indicator: $\hat{f}(X) = \mathbf{1}(\hat{\tau}(x) + b - a \geq 0)$ using the first part of the data. Let $\hat{\eta}$ be the set of all nuisance estimates, including now $\hat{f}(X)$.
4: Find $\hat{\beta}$ that solves the following estimating equation on $\mathcal{D}_2$: $\frac{1}{|\mathcal{D}_2|} \sum_{X \in \mathcal{D}_2} \varphi(X, \hat{\beta}, \hat{\eta}) = 0$ where $\varphi(X, \hat{\beta}, \hat{\eta})$ is constructed using $\hat{\eta}$ following the expression in Theorem 1.
5: Output $\hat{\beta}$

---

### 3.4 Bounds in a sensitivity model

In some cases, we may be willing to impose additional assumptions limiting the deviation between the restricted and fully conditional CATEs. This may be justified based either on domain knowledge, or taken in the spirit of a sensitivity analysis where the analyst varies a parameter controlling the strength of such assumptions to see how much variation across levels of $W$ their conclusions are robust to. In this section, we propose such a sensitivity analysis model to formalize the case where $W$ is believed to have a limited impact on treatment effects, after conditioning on $V$. Specifically, our sensitivity model imposes the assumption that

$$\left| \mathbb{E}(Y^1 - Y^0 \mid V = v, W = w, E = 1) - \mathbb{E}(Y^1 - Y^0 \mid V = v, E = 1) \right| \leq \delta \tag{5}$$

or that the fully conditional effects cannot differ from partly conditional effects by more than $\delta$. $\delta$ here is a user-chosen parameter, that may be varied to test the robustness of estimates to an increasingly strong effect of $W$. At $\delta = 0$, the fully conditional CATE is equal to the restricted CATE, and at $\delta = b - a$, we recover our previous bounds (that use only the boundedness of the outcome to $[a, b]$). At any intermediate level of $\delta$, we obtain partial identification bounds that are of a similar form to those in Theorem 1 (but stronger), simply replacing the terms $(a - b)$ or $(b - a)$ by $\mathbb{E}(Y^1 - Y^0 \mid V = v, E = 1) \pm \delta$. We estimate these bounds using the same strategy as for the original model, plugging in the identified quantity $\mathbb{E}(Y^1 - Y^0 \mid V = v, E = 1) = \mu_1(v) - \mu_0(v)$ using data from the study population.

## 4 Experiments

**Setup:** We start our experimentation with experiments on simulated data so that the ground-truth CATEs are known, and afterwards give an application on data from a real RCT. Full details of the simulation are given in Appendix. The process samples 10,000 observations of 3 continuous covariates for $V$ and 3 discrete (binary) covariates for $W$. $W$ is simulated as $\mathbb{P}(W_i = 1) = \text{logit}(\alpha_i^T V)$ for a coefficient vector $\alpha_i$, where the choice of $\alpha$ allows us to control the degree of dependence between $V$ and $W$. $E$ and $T$ are generated similarly as functions of $V$, producing covariate shift between the populations and nonuniform assignment to treatment across levels of $V$ within the study population. Finally, the CATE is a specified function of both $V$ and $W$ and we sample observed outcomes from the study population matching the CATE.

**Baselines:** To our knowledge, ours is the first paper to present an algorithm to identify and estimate CATE bounds for covariates unobserved in the original study. We compare to three baselines. First, we compare the performance of our bias-correct estimator to the plug-in estimator discussed in Section 3.2, which uses the same nuisance function estimates as our model but estimates $\beta$ directly by plugging these estimates into the moment condition, without the influence function-based correction. Second, we compare the informativeness of our bounds to those which use all of our assumptions *except* the key ecological inference component in Equation 1, to test whether using this information results in tighter identification (more details on this below). Third, we compare the frequency with which our bounds cover the true treatment effects for specific subgroups of $(V, W)$ with the coverage of confidence intervals for the restricted CATE (which is measured on $V$ and not $W$).

**Benchmarking:** In order to set the sensitivity parameter $\delta$, we can use benchmarking methods similar to other sensitivity models in causal inference. (Byun et al., 2024; Hosman et al., 2010; McClean et al., 2024). For a specific instance, the analyst holds out n variables from $V$ to form a "simulated" set of additional covariates $W'$ mimicking $W$ where $V'$ represents the original $V$ excluding $W'$. They then use the difference $|\mathbb{E}[Y^1 - Y^0|V'] - \mathbb{E}[Y^1 - Y^0|V', W']|$ as a proxy for $|\mathbb{E}[Y^1 - Y^0|V] - \mathbb{E}[Y^1 - Y^0|V, W]|$ in order to set $\hat{\delta}$. This procedure is performed on all subsets of n variables in $V$ and results are averaged. As in sensitivity analyses for unmeasured confounding, this provides an interpretation (based on domain knowledge) that the analysis is robust to variables "at least as important" as $W'$.

**Simulation:** Figure 2a demonstrates the difference in estimation error between the plug in and bias corrected estimators across 200 random seeds where we inject varying amounts of error into estimates of outcome and propensity models. We find that the bias-corrected model produces more accurate estimates of the bounds (known in simulation) than the plug-in by large margins in the vast majority of situations, especially when there is a high amount of error in the outcome regression. The plug-in model performs better only when there is little error in outcome regression modeling and high error in propensity modeling. Intuitively, we might expect ecological bounds to be tighter when the $V$ is more informative about $W$. To test this, we vary the distribution of $\nu$ and plot its entropy against the width of our bias-corrected bounds. We observe that as the entropy decreases, the average bound size also decreases (Figure 2b). This shows that when the known covariates in the study population become more predictive of the new covariates in the target population, our bounds are tighter.

In Figure 3a, we apply the sensitivity model discussed in section 3.4. The left plot shows the average value of the bounds as a function of the sensitivity parameter $\delta$. As expected, the CATE is point-identified at $\delta = 0$, with progressively greater uncertainty as $\delta$ grows. For reference, we also plot bounds (in black) which use *only* the sensitivity assumption in Equation 5. By itself, this assumption implies bounds of width $2\delta$ for the CATE. The bounds output by our method are substantially stronger (less than half of the width), indicating that our ecological inference framework which uses the joint covariate distribution provides substantially more informative inferences than would otherwise be possible. The right figure shows the rate at which our bounds cover the true CATE as a function of $\delta$. As expected, if $\delta$ is close to its true value (known in the simulation), we observe close to nominal coverage levels. Additionally, Figure 3b shows that the 95% CI of the restricted CATE (measured on $V$ and not $W$) baseline only correctly covers 43.7 percent of the true treatment effects, while at both the true and estimated values of our sensitivity parameter $\delta$ ($\hat{\delta}$ comes from Benchmarking distribution in Figure 2c, estimated and true $\delta$ are very close - within 10% of each other), roughly 98 percent of the true treatment effects are covered by our method. We can also observe that at

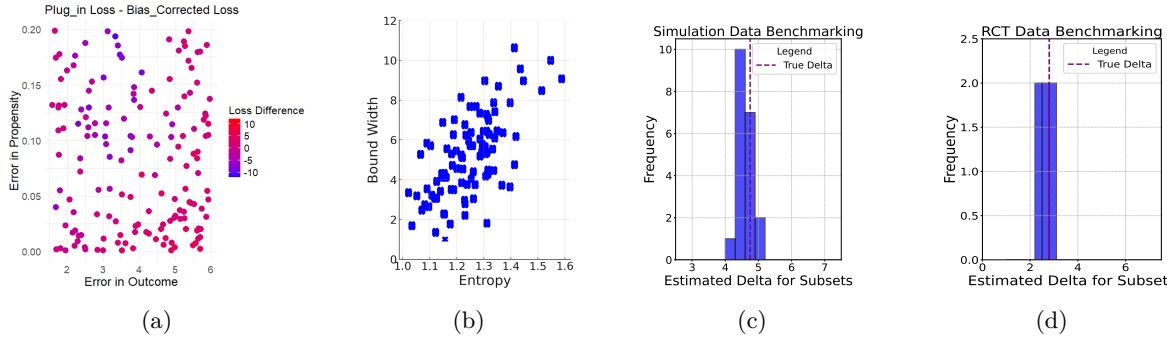

(a)  (b)  (c)  (d)

Figure 2: (a) Difference in estimation error between the plug in and bias corrected estimators under varying errors in outcome and propensity modeling. Red is higher loss for the plug-in. All errors are measured in mean absolute deviation. (b) Average worst-case bound width as a function of the entropy of $\nu$ where the range of outcomes is 40. (c,d) Benchmarking distribution for (c): Simulation Data, (d) RCT Data. Final $\hat{\delta}$ represents the mean of each distribution and is within $\sim 10\%$ of both true $\delta$. We consider all potential subsets of V' and W' that can be made from V given the number of variables in W' (3 in Simulation and 1 in RCT). Full range of possible $\delta$ for x-axis is (0,16).

both the true and estimated values of $\delta$, the bounds are very informative as the CI of the mean lower bound of the treatment effect ($\gamma_l(v, w)$) exceeds 0. More specifically, we have that $\gamma_l(v, w) \sim \in (1.1, 1.6)$ at $\hat{\delta}$.

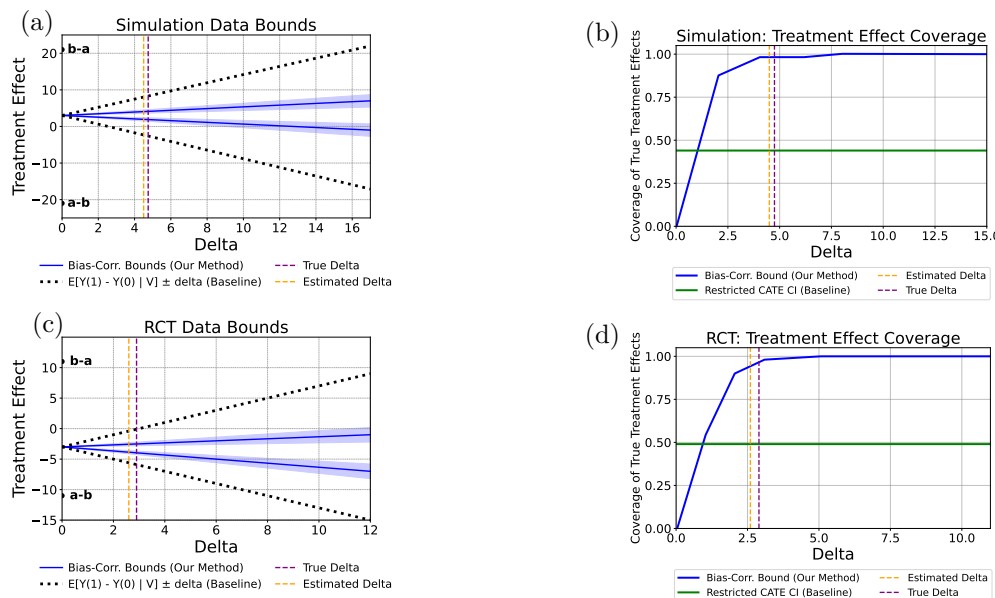

Figure 3: Left (a,c): Bounds in the sensitivity model as a function of $\delta$, averaged over units. The blue lines are bounds output by our method (Bias Correction), with a 95% CI. The black lines are bounds that use only the sensitivity assumption in Equation 5 without our ecological inference framework. Right (b,d): Blue line is the fraction of times our bounds cover the true CATE compared to green line which is the baseline coverage resulting from a Doubly Robust estimate of the Restricted CATE's 95 % CI. Top(a,b): Simulation Data. Bottom(b,c): RCT Data.

**Real World Data:** Finally, we illustrate our method by estimating conditional treatment effects for a real-life RCT that measured the effect of a throat treatment (gargling with a licorice solution) prior to thoracic surgery on post-operative swallowing pain (Ruetzler et al., 2013). We split the dataset into a "study" and "target" population by holding out one of the covariates in the study population to form $W$. All of

the variables in *V* are standard measurements such as pain index, while the variable we separated for *W* represents an indicator as to whether the patient experienced coughing after the ventilation tube was removed from their airway. Although this indicator is considered by many physicians to be a valuable metric (and would be visible to physicians working with the target population), it is frequently not recorded (Duan et al., 2021) and hence might not be available in a study dataset. Further details about variables and the setting of the RCT can be found in the Appendix. Figure 3c shows the average width of bounds we obtain as a function of $\delta$, and we once again observe that the ecological framework provides substantially improved identifying power compared to naive bounds that only use the sensitivity assumption itself.

We again see very informative bounds in Figure 3c. In particular, the CI of the mean upper bound of the treatment effect lies entirely below 0 ($\gamma_u(v, w) \sim\in (-2.4, -3.0)$) at $\hat{\delta}$ where $\hat{\delta}$ comes from the Benchmarking process shown in Figure 2d, indicating a provable reduction in postoperative throat pain due to the treatment. At the estimated value of $\delta$, our bounds have almost exactly the desired 95% coverage level (Figure 3d). By contrast, the bounds output by a DR estimator for the restricted CATE have only 50% coverage. This indicates that the held-out covariate has a significant impact on treatment effects, such that the expected effect for many patients moves outside the original CI after seeing the new covariate. Our bounds properly account for this uncertainty while remaining informative about effects. Similarly to the simulated data, even if we select a value of $\delta$ that matches the average width of CIs for the restricted CATE, our bounds have 25% better coverage. This shows how leveraging the joint covariate distribution allows us to provide strictly more informative uncertainty quantification about treatment effects than otherwise would be possible.

## 5    Discussion

In this paper, we give the first formal presentation of an identification and estimation strategy for the CATE when generalizing to a target population that has covariates not observed in an earlier study. We develop a bias-corrected estimator that retains fast $O_{\mathbb{P}}(\frac{1}{\sqrt{n}})$ convergence rates even when nuisances are estimated nonparametrically, and is asymptotically normal under standard conditions. We also introduce a sensitivity model to bound impact of the new covariates on treatment effects. Empirically, we find that our method is often able to substantially reduce uncertainty about heterogeneous treatment effects, even in this challenging setting where no outcome data directly linked to the new covariates is observed. We caution that, like all causal inference methods, our framework requires domain expertise to assess the plausibility of assumptions, e.g. that *W* and *V* follow a consistent joint distribution between the populations. However, when used appropriately, our framework gives users one way to assess the generalizability of effect estimates to newly identified subpopulations *before* committing to a treatment assignment policy, helping to avert unintended negative consequences.

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

## A Appendix

In this appendix we provide full derivations of the bounds, moment conditions, and influence-function-based expansions used in the main text. We also add explanatory commentary to make the main algebraic steps more explicit.

Throughout, we keep the same notation as in the main body. In particular, we use:

- $\mathbb{E}[\cdot]$ to denote expectations.

- $\mathbb{P}(\cdot)$ to denote probabilities.

- $E \in \{0, 1\}$ exclusively as the indicator of the study/target population (not as an expectation operator).

### A.1 Fully Conditional Bounds

We begin by deriving bounds on the conditional mean $\mathbb{E}(Y \mid W = w, V = v)$ when we only know that $Y$ lies in a bounded interval and we observe the distribution of $(Y, W, V)$.

Assume that $Y \in [a, b]$ with probability one. We first apply the law of total expectation by conditioning on the event $\{W = w\}$ vs. $\{W \neq w\}$:

$$\mathbb{E}(Y \mid W = w, V = v) = \frac{\mathbb{E}(Y \mid V = v) - \mathbb{E}(Y \mid W \neq w, V = v)\mathbb{P}(W \neq w \mid V = v)}{\mathbb{P}(W = w \mid V = v)}.$$

The expression above is an identity that simply rewrites $\mathbb{E}(Y \mid V = v)$ in terms of $\mathbb{E}(Y \mid W = w, V = v)$ and $\mathbb{E}(Y \mid W \neq w, V = v)$, and then solves for $\mathbb{E}(Y \mid W = w, V = v)$.

Because $Y \in [a, b]$ almost surely, we know that $\mathbb{E}(Y \mid W \neq w, V = v)$ must lie in $[a, b]$. Minimizing and maximizing the right-hand side over all such values yields the sharp interval:

$$\mathbb{E}(Y \mid W = w, V = v) \in \left[ \max\left\{ \frac{\mathbb{E}(Y \mid V = v) - b\mathbb{P}(W \neq w \mid V = v)}{\mathbb{P}(W = w \mid V = v)}, a \right\}, \right.$$
$$\left. \min\left\{ \frac{\mathbb{E}(Y \mid V = v) - a\mathbb{P}(W \neq w \mid V = v)}{\mathbb{P}(W = w \mid V = v)}, b \right\} \right].$$

We now apply the same idea to the conditional average treatment effect $Y^1 - Y^0$ instead of $Y$. Under the assumptions laid out in the Problem Setup (in particular, that treatment is randomized within the experimental sample), we can write:

$$\mathbb{E}(Y^1 - Y^0 \mid V = v, W = w, E = 1)$$
$$= \frac{\mathbb{E}(Y^1 - Y^0 \mid V = v, E = 1) - \mathbb{E}(Y^1 - Y^0 \mid W \neq w, V = v, E = 1)\mathbb{P}(W \neq w \mid V = v, E = 0)}{\mathbb{P}(W = w \mid V = v, E = 0)}$$
$$= \frac{\mathbb{E}(Y \mid V = v, T = 1, E = 1) - \mathbb{E}(Y \mid V = v, T = 0, E = 1) - \alpha(v, w)\mathbb{P}(W \neq w \mid V = v, E = 0)}{\mathbb{P}(W = w \mid V = v, E = 0)}.$$

The first equality above again uses the law of iterated expectations, now applied to $Y^1 - Y^0$ across the partition $\{W = w\}$ vs. $\{W \neq w\}$. The second equality uses randomization of $T$ in the experiment to replace $\mathbb{E}(Y^t \mid V = v, E = 1)$ by the observed regression $\mathbb{E}(Y \mid V = v, T = t, E = 1)$, and defines

$$\alpha(v, w) = \mathbb{E}(Y^1 - Y^0 \mid W \neq w, V = v, E = 1).$$

In this representation the only unknown quantity at $(v, w)$ is the conditional effect

$$\gamma(v, w) := \mathbb{E}(Y^1 - Y^0 \mid V = v, W = w, E = 1).$$

If all that is known is that $Y \in [a, b]$, then (by monotonicity) we have $Y^1 - Y^0 \in [a - b, b - a]$ almost surely. This implies that the nuisance term $\alpha(v, w)$ is only known to lie in $[a - b, b - a]$ pointwise in $(v, w)$.

Plugging the extremal values of $\alpha(v, w)$ into the identity above, and combining this with the bounds for $\mathbb{E}(Y \mid W = w, V = v)$, yields sharp bounds

$$\gamma_\ell(v, w) \leq \mathbb{E}(Y^1 - Y^0 \mid V = v, W = w, E = 1) \leq \gamma_u(v, w),$$

where

$$\gamma_\ell(v, w) = \max \left\{ \frac{\mu_1(v) - \mu_0(v) - (b - a)\mathbb{P}(W \neq w \mid V = v, E = 0)}{\mathbb{P}(W = w \mid V = v, E = 0)}, (a - b) \right\}$$

$$\gamma_u(v, w) = \min \left\{ \frac{\mu_1(v) - \mu_0(v) - (a - b)\mathbb{P}(W \neq w \mid V = v, E = 0)}{\mathbb{P}(W = w \mid V = v, E = 0)}, (b - a) \right\},$$

and where we have defined the experimental outcome regressions

$$\mu_t(v) = \mathbb{E}(Y \mid V = v, T = t, E = 1).$$

## A.2 Moment Condition

We next make explicit how the projection moment condition $M(\beta)$ looks when we plug in the truncated lower and upper bound functions. The idea is simply that we regress a (possibly truncated) bound function onto a working model $m(v, w; \beta)$ using instruments $g(x)$.

For the lower bound, recall that we define $\gamma_\ell(v, w) = \max\{\tau_\ell(x), a - b\}$, where $x = (v, w)$. Then the population moment condition is

$$M(\beta) = \mathbb{E}[g(x)(\gamma(v, w) - m(v, w; \beta))]$$
$$= \mathbb{E}[g(x)(\max\{\tau_\ell(x), a - b\} - m(v, w; \beta))].$$

We can rewrite the maximum explicitly in terms of indicator functions:

$$M(\beta) = \mathbb{E}[g(x)\{\tau_\ell(x) * 1(\tau_\ell(x) \geq (a - b)) + (a - b) * 1(\tau_\ell(x) \leq (a - b)) - m(v, w; \beta)\}]$$
$$= \mathbb{E}[g(x)\{(\tau_\ell(x) + b - a) * 1(\tau_\ell(x) + b - a \geq 0) + a - b - m(v, w; \beta)\}].$$

The last line is just a reparameterization of the indicator in terms of $\tau_\ell(x) + b - a$, which will be convenient when we apply the margin condition.

For the upper bound, we truncate at $b - a$ instead of $a - b$. Let $\gamma_u(v, w) = \min\{\tau_u(x), b - a\}$. Then

$$M(\beta) = \mathbb{E}[g(x)(\gamma(v, w) - m(v, w; \beta))]$$
$$= \mathbb{E}[g(x)(\min\{\tau_u(x), b - a\} - m(v, w; \beta))]$$
$$= \mathbb{E}[g(x)\{(\tau_u(x) + a - b) * 1(\tau_u(x) + a - b \leq 0) + b - a - m(v, w; \beta)\}].$$

Thus, for both lower and upper bounds, the moment condition has the same form: a weighted residual of a truncated function of $\tau_\ell(x)$ or $\tau_u(x)$ against the working model $m(v, w; \beta)$.

## A.3 Use of Margin Condition

We now record the basic inequality that underlies the use of the margin condition. This inequality controls the error incurred when we replace the unknown sign of $\tau(x)$ by the sign of an estimator $\hat{\tau}(x)$ inside indicator functions.

Consider the difference

$$\mathbb{E}[\tau \mathbf{1}(\hat{\tau} + b - a) - \tau \mathbf{1}(\tau + b - a)] = \mathbb{E}[\tau \mathbf{1}(\hat{\tau}) - \tau \mathbf{1}(\tau)].$$

We can bound the absolute value of this difference using the triangle inequality:

$$\mathbb{E}[\tau \mathbf{1}(\hat{\tau}) - \tau \mathbf{1}(\tau)] \leq \mathbb{E}[|\tau| \cdot |\mathbf{1}(\hat{\tau}) - \mathbf{1}(\tau)|].$$

The indicator difference is only nonzero where $\hat{\tau}$ and $\tau$ have different signs; this in turn implies that $|\tau| \leq |\hat{\tau} - \tau|$ on that event. Using this observation, we obtain

$$
\begin{aligned}
\mathbb{E}[|\tau| \cdot |\mathbf{1}(\hat{\tau}) - \mathbf{1}(\tau)|] &\leq \int |\hat{\tau} - \tau| \mathbf{1}(|\tau| \leq |\hat{\tau} - \tau|) dP(x) \\
&\leq \int |\hat{\tau} - \tau| \mathbf{1}(|\tau| \leq |\hat{\tau} - \tau|) dP(x) \\
&\leq \sup_x |\hat{\tau} - \tau| \cdot P(|\tau| \leq ||\hat{\tau} - \tau||_\infty) \\
&\leq C ||\hat{\tau} - \tau||_\infty^{1+\alpha}.
\end{aligned}
$$

The third line follows from the fact just mentioned: if $\tau$ and $\hat{\tau}$ are of opposite signs, then $|\tau| \leq |\hat{\tau} - \tau|$. The last inequality is exactly where the margin condition is invoked; it controls the probability mass near zero and yields a $(1 + \alpha)$-order bound in the sup-norm error.

## A.4   Influence Function Derivation

We now derive the influence function used in the main text. The strategy is to start from basic, well-known influence functions and then apply the usual "influence function calculus" (e.g., product and quotient rules) to obtain the influence function of the more complicated functional of interest.

**Basic facts.**   We first collect several standard influence functions and identities that we will use as building blocks; see Section 3.4.3 of Kennedy (2023) for more background and proofs:

$$
\begin{aligned}
IF(\mathbb{E}(Y \mid X = x)) &= \frac{1(X = x)}{P(X = x)} \{Y - \mathbb{E}[Y \mid X = x]\}, \\
P(W = w \mid X = x) &= \mathbb{E}[1(W = w) \mid X = x], \\
\frac{p(x)}{p(V = v, T = 1, E = 1)} &= \frac{p(W = w \mid V = v)}{p(T = 1, E = 1 \mid V = v)}, \\
IF(p(x)) &= \{1(X = x) - p(x)\}.
\end{aligned}
$$

Here $IF(\cdot)$ denotes the influence function of the corresponding functional, and $p(\cdot)$ is the density/mass function of $X$.

**Derivation.**   We now apply these basic components to our particular ratio functional. For clarity, we introduce explicit notation for the numerator and denominator of the scalar quantity $\tau(x)$.

We say that $\tau_{\text{num}}$ is the numerator of $\tau$,

$$
\tau_{\text{num}}(x) = \mu_1(v) - \mu_0(v) - (b - a)\mathbb{P}(W \neq w \mid V = v, E = 0),
$$

and that $\tau_{\text{den}}$ is the denominator of $\tau$,

$$
\tau_{\text{den}}(x) = \mathbb{P}(W = w \mid V = v, E = 0),
$$

so that $\tau(x) = \tau_{\text{num}}(x)/\tau_{\text{den}}(x)$. We use "num/den" here to avoid the subscript $n$ being confused with a sample size.

As mentioned in the paper, we also define the indicator

$$
f(x) = 1(\tau_\ell(x) + b - a \geq 0).
$$

Using the quotient rule for influence functions, we obtain

$$
IF(\tau(x)) = IF\left(\frac{\tau_{\text{num}}(x)}{\tau_{\text{den}}(x)}\right)
$$

$$= \frac{IF(\tau_{\text{num}}(x))}{\tau_{\text{den}}(x)} - \frac{\tau_{\text{num}}(x)}{\tau_{\text{den}}(x)^2}IF(\tau_{\text{den}}(x)).$$

We now spell out the influence functions of the numerator and denominator in terms of the nuisance functions involved:

$$IF(\tau_{\text{num}}(x)) = \frac{ET*1(V=v)}{p(V=v,T=1,E=1)}(Y-\mu_1(v)) - \frac{E(1-T)*1(V=v)}{p(V=v,T=0,E=1)}(Y-\mu_0(v))$$

$$+ \frac{(b-a)1(V=v,E=0)}{p(V=v,E=0)}\{1(W=w)-p(W=w\mid V=v,E=0)\},$$

$$IF(\tau_{\text{den}}(x)) = \frac{1(V=v,E=0)}{P(V=v,E=0)}\{1(W=w)-P(W=w\mid V=v,E=0)\}.$$

Here $ET1(V=v)$ is the indicator of the event $\{E=1,T=1,V=v\}$, and the expressions follow from the standard influence functions of conditional means and conditional probabilities as in the "facts" section above.

We now consider the functional

$$\sum_x g(x)\{\tau(x)f(x)-q(x)\}p(x),$$

where we call the known portion $q(x)$ such that

$$q(x) = -((b-a)f(x)+a-b-m(x;\beta)).$$

Applying the product and chain rules for influence functions to this functional yields

$$IF\left[\sum_x g(x)\{\tau(x)f(x)-q(x)\}p(x)\right]$$

$$= \left[\sum_x \left(g(x)\{IF(\tau(x))f(x)\}p(x) + g(x)\{\tau(x)f(x)-q(x)\}IF(p(x)))\right)\right]$$

$$= \left[\sum_x \left(g(x)\left\{\frac{IF(\tau_{\text{num}}(x))}{\tau_{\text{den}}(x)} - \frac{\tau(x)IF(\tau_{\text{den}}(x))}{\tau_{\text{den}}(x)}\right\}f(x)p(x)\right) + g(X)\{\tau(X)f(X)-q(X)\}\right].$$

For convenience, we decompose this influence function into three pieces:

$$\varphi_1(X;\beta) = \sum_x g(x)\left\{\frac{IF(\tau_{\text{num}}(x))}{\tau_{\text{den}}(x)}\right\}f(x)p(x),$$

$$\varphi_2(X;\beta) = -\sum_x g(x)\left\{\frac{\tau(x)IF(\tau_{\text{den}}(x))}{\tau_{\text{den}}(x)}\right\}f(x)p(x),$$

$$\varphi_3(X;\beta) = g(x)\{\tau(x)f(x)-q(x)\}.$$

Then the overall influence function is

$$\varphi(X;\beta) = \varphi_1(X;\beta) + \varphi_2(X;\beta) + \varphi_3(X;\beta).$$

We now simplify $\varphi_1$ and $\varphi_2$ by substituting the concrete forms of $IF(\tau_{\text{num}}(x))$ and $IF(\tau_{\text{den}}(x))$, and using the assumptions on $p(W\mid V)$ (e.g., that it is common across $E$). After straightforward algebra, we obtain:

$$\varphi_1(X;\beta) = \frac{ET}{p(T=1,E=1\mid V)}(Y-\hat{\mu}_1(V))*\sum_w g(V,w)f(V,w)p(w\mid V)*\frac{1}{\hat{\tau}_{\text{den}}(V,w)}$$

$$- \frac{E(1-T)}{p(T=0,E=1\mid V)}(Y-\hat{\mu}_0(V))*\sum_w g(V,w)f(V,w)p(w\mid V)*\frac{1}{\hat{\tau}_{\text{den}}(V,w)}$$

$$+ \frac{(b-a)(1-E)}{p(E=0\mid V)}\left\{g(V,W)f(V,W)*\frac{p(W\mid V)}{\hat{\tau}_{\text{den}}(V,W)}\right.$$

$$-\sum_{w} g(V,w)f(V,w)p(w \mid V) * \frac{p(W = w \mid V, E = 0)}{\hat{\tau}_{\text{den}}(V,w)} \Bigg\}$$

$$= \frac{ET}{p(T = 1, E = 1 \mid V)}(Y - \hat{\mu}_1(V)) * \sum_{w} g(V,w)f(V,w)$$

$$- \frac{E(1 - T)}{p(T = 0, E = 1 \mid V)}(Y - \hat{\mu}_0(V)) * \sum_{w} g(V,w)f(V,w)$$

$$+ \frac{(b - a)(1 - E)}{p(E = 0 \mid V)}\Big\{ g(V,W)f(V,W) - \sum_{w} g(V,w)f(V,w)p(w \mid V) \Big\}.$$

To go from the first expression for $\varphi_1$ to the second, we use that $\hat{\tau}_{\text{den}}(V,W) = \hat{p}(W \mid V)$ (by construction) and that, under our assumptions, $\hat{p}(W \mid V)$ does not depend on $E$.

Similarly, for $\varphi_2$ we obtain

$$\varphi_2(X; \beta) = -\frac{1 - E}{\hat{p}(E = 0 \mid V)}\Big\{ g(V,W)f(V,W)\hat{p}(W \mid V) * \frac{\hat{\tau}(X)}{\hat{\tau}_{\text{den}}(X)}$$

$$- \sum_{w} g(V,w)f(V,w)\hat{p}(w \mid V) * \frac{\hat{\tau}(V,w)\hat{p}(W = w \mid V, E = 0)}{\hat{\tau}_{\text{den}}(V,w)} \Big\}$$

$$= -\frac{1 - E}{\hat{p}(E = 0 \mid V)}\Big\{ g(V,W)f(V,W)\hat{\tau}(X)$$

$$- \sum_{w} g(V,w)f(V,w)\hat{\tau}(V,w)\hat{p}(W = w \mid V, E = 0) \Big\}.$$

Putting these pieces together, we have

$$\varphi(X; \beta) = \varphi_1(X; \beta) + \varphi_2(X; \beta) + \varphi_3(X; \beta),$$

and the result stated in the paper follows upon specifying the choice $g(x) = [v^\top, w^\top]^\top = x$.

Writing $\varphi(X; \beta)$ explicitly, we obtain

$$\varphi(X; \beta) = \frac{ET}{p(T = 1, E = 1 \mid V)}(Y - \hat{\mu}_1(V)) * \sum_{w} g(V,w)f(V,w)$$

$$- \frac{E(1 - T)}{p(T = 0, E = 1 \mid V)}(Y - \hat{\mu}_0(V)) * \sum_{w} g(V,w)f(V,w)$$

$$+ \frac{(b - a)(1 - E)}{p(E = 0 \mid V)}\Big\{ g(V,W)f(V,W) - \sum_{w} g(V,w)f(V,w)p(w \mid V) \Big\}$$

$$- \frac{1 - E}{\hat{p}(E = 0 \mid V)}\Big\{ g(V,W)f(V,W)\hat{\tau}(X)$$

$$- \sum_{w} g(V,w)f(V,w)\hat{\tau}(V,w)\hat{p}(W = w \mid V, E = 0) \Big\}$$

$$+ g(X)\{\tau_\ell(X)f(X) + (b - a)f(X) + a - b - m(X, \beta)\}.$$

The influence function for the upper bound $\gamma_u(x)$ has an identical structure, with the truncation endpoints swapped. Specifically,

$$\varphi(X; \beta) = \frac{ET}{p(T = 1, E = 1 \mid V)}(Y - \hat{\mu}_1(V)) * \sum_{w} g(V,w)f(V,w)$$

$$- \frac{E(1 - T)}{p(T = 0, E = 1 \mid V)}(Y - \hat{\mu}_0(V)) * \sum_{w} g(V,w)f(V,w)$$

$$+ \frac{(b-a)(1-E)}{p(E=0 \mid V)}\Big\{ g(V,W)f(V,W) - \sum_w g(V,w)f(V,w)p(w \mid V)\Big\}$$

$$- \frac{1-E}{\hat{p}(E=0 \mid V)}\Big\{ g(V,W)f(V,W)\hat{\tau}(X)$$

$$- \sum_w g(V,w)f(V,w)\hat{\tau}(V,w)\hat{p}(W=w \mid V, E=0)\Big\}$$

$$+ g(X)\{\tau_\ell(X)f(X) + (a-b)f(X) + a - b - m(X,\beta)\},$$

where, for the upper bound, we use $f(X) = 1(\tau_u(x) + a - b \leq 0)$ and flip $a-b$ and $b-a$ wherever they occur.

### A.4.1  Bias Derivation

We now turn to the bias calculation underlying the second-order remainder term. In this subsection we keep the algebra as originally written, and insert commentary describing the main steps and tools (iterated expectation, algebraic rearrangement, and grouping of terms into products of nuisance estimation errors).

In the steps below, we primarily use iterated expectation and algebraic manipulation. For notation purposes we say $P(\bar{W} \mid V) = 1 - P(W \mid V)$. We will eventually convert some of the nuisance functions to the notation used in the paper (e.g. $\nu(V,w) = p(W=w \mid V, E=0)$ and $\rho_0(V) = p(E=0 \mid V)$), but for now we leave them in their original form as it makes them easier to visualize and manipulate. For brevity we use $\pi_x = p(T=x, E=1 \mid V)$ in these proofs. It is also important to recall that $\hat{\tau}_{\mathrm{den}}(V,W) = \hat{p}(W \mid V)$, so we can interchange these as we like.

Our general strategy is to write the bias as an expectation of the difference between the estimated and "true" influence functions, and then algebraically rearrange this expression so that every term is either second order or can be decomposed into a sum of a first-order and a second-order term. We focus first on rewriting each candidate first-order term as a sum of a second-order term plus a remainder, and then handle any remaining first-order pieces.

The bias is given by:

$$R_n = \mathbb{P}\{\varphi(X;\beta,\hat{\eta}) - \varphi(X;\beta,\eta_0)\} = \mathbb{P}\{\varphi(X;\beta,\hat{\eta})\}.$$

We start by examining the contribution from $\varphi_1$:

$$\mathbb{E}\{\varphi_1(X;\beta,\hat{\eta})\} = \mathbb{E}\Big[ \frac{ET}{\hat{p}(T=1, E=1 \mid V)}(Y - \hat{\mu}_1(v)) * \sum_x g(V,w)f(V,w)$$

$$- \frac{E(1-T)}{\hat{p}(T=0, E=1 \mid V)}(Y - \hat{\mu}_0(v)) * \sum_x g(V,w)f(V,w)$$

$$+ \frac{(b-a)(1-E)}{\hat{p}(E=0 \mid V)}\{g(X)f(X) - \sum_w g(V,w)f(V,w)\hat{p}(w \mid V)\}\Big]$$

$$= \mathbb{E}\Big[ \frac{\pi_1}{\hat{\pi}_1}(\mu_1(V) - \hat{\mu}_1(V)) * \sum_x g(V,w)f(V,w)$$

$$- \frac{\pi_0}{\hat{\pi}_0}(\mu_0(V) - \hat{\mu}_0(V)) * \sum_x g(V,w)f(V,w)$$

$$+ \frac{(b-a)p(E=0 \mid X)}{\hat{p}(E=0 \mid V)}g(X)f(X)$$

$$- \frac{(b-a)p(E=0 \mid V)}{\hat{p}(E=0 \mid V)}\sum_w g(V,w)f(V,w)\hat{p}(w \mid V)\Big].$$

In the last equality, we have taken conditional expectations given $V$ and used the definitions of $\pi_x$ and the various nuisance functions. We then expand each ratio, such as $\pi_1/\hat{\pi}_1$, around 1 to separate first-order and second-order parts.

Continuing this expansion, we write:

$$= \mathbb{E}\big[\frac{\pi_1 - \hat{\pi}_1}{\hat{\pi}_1}(\mu_1(V) - \hat{\mu}_1(V)) * \sum_x g(V, w)f(V, w)$$

$$- \frac{\pi_0 - \hat{\pi}_0}{\hat{\pi}_0}(\mu_0(V) - \hat{\mu}_0(V)) * \sum_x g(V, w)f(V, w)\hat{p}(w \mid V)$$

$$+ (\mu_1(V) - \hat{\mu}_1(V)) * \sum_x g(V, w)f(V, w)$$

$$- (\mu_0(V) - \hat{\mu}_0(V)) * \sum_x g(V, w)f(V, w)$$

$$+ \frac{(b-a)p(E=0 \mid X)}{\hat{p}(E=0 \mid V)}g(X)f(X) - \frac{(b-a)p(E=0 \mid V)}{\hat{p}(E=0 \mid V)} \sum_w g(V, w)f(V, w)\hat{p}(w \mid V)\big].$$

The terms involving $(\pi_x - \hat{\pi}_x)(\mu_x - \hat{\mu}_x)$ are already second order. The remaining two terms involving $p(E=0 \mid \cdot)$ are the most delicate and are handled next.

We now isolate and manipulate those latter terms:

$$\mathbb{E}\big[(b-a) \sum_x \frac{p(E=0 \mid x)}{\hat{p}(E=0 \mid v)}g(x)f(x)p(w \mid v)p(v)\big]$$

$$- (b-a) \sum_x \frac{p(E=0 \mid v)}{\hat{p}(E=0 \mid v)}g(x)f(x)\hat{p}(w \mid v)p(V)$$

$$= \mathbb{E}\big[(b-a) \sum_x \frac{p(E=0 \mid x)}{\hat{p}(E=0 \mid v)}g(x)f(x)p(w \mid v)p(v)$$

$$- (b-a) \sum_x \frac{p(E=0 \mid v)}{\hat{p}(E=0 \mid v)}g(x)f(x)\hat{p}(w \mid v)p(v)\big]$$

$$= \mathbb{E}\big[(b-a) \sum_x g(x)f(x)p(v)\{\frac{p(w \mid v)p(E=0 \mid x)}{\hat{p}(E=0 \mid v)} - \frac{\hat{p}(W=w \mid V, E=0)p(E=0 \mid v)}{\hat{p}(E=0 \mid v)}\}\big]$$

$$= \mathbb{E}\big[(b-a) \sum_x g(x)f(x)p(v)\{\frac{p(E=0 \mid v)}{\hat{p}(E=0 \mid v)} - 1\}*$$

$$\{p(W=w \mid V=v, E=0) - \hat{p}(W=w \mid V=v, E=0)\}$$

$$+ (b-a) \sum_x g(x)f(x)\{p(W=w \mid V=v, E=0) - \hat{p}(W=w \mid V=v, E=0)\}\big]$$

$$= \mathbb{E}\big[(b-a) \sum_w g(V, w)f(V, w)\{\frac{p(E=0 \mid V)}{\hat{p}(E=0 \mid V)} - 1\}\{p(W=w \mid V, E=0) - \hat{p}(W=w \mid V, E=0)\}\big]$$

$$a + \mathbb{E}\big[(b-a) \sum_w g(V, w)f(V, w)\{p(W=w \mid V, E=0) - \hat{p}(W=w \mid V, E=0)\}\big].$$

This algebra rearranges the terms so that we can clearly see (i) a second-order product of errors in $p(E=0 \mid V)$ and $p(W=w \mid V, E=0)$, and (ii) a leftover first-order term in $p(W=w \mid V, E=0)$. The leftover term will later be paired with pieces from $\varphi_2$ and $\varphi_3$ and turned into a second-order term as well.

Next, we handle the remaining first-order terms in $\varphi_1$ together with the contributions from $\varphi_2$ and $\varphi_3$. For bookkeeping, it is convenient to rewrite $\varphi$ as

$$\varphi = \varphi_1(X; \beta) + \varphi_2(X; \beta) + g(X)\left(\frac{\hat{\tau}_{\text{num}}(X)}{\hat{\tau}_{\text{den}}(X)}f - q(X)\right) - g(X)\left(\frac{\tau_{\text{num}}}{\tau_{\text{den}}(X)}f - q(X)\right)$$

$$= \varphi_2(X; \beta) + g(X)\left(\frac{\hat{\tau}_{\text{num}}(X)}{\hat{\tau}_{\text{den}}}f - q(X)\right) - g(X)\left(\frac{\hat{\tau}_{\text{num}}(X)}{\tau_{\text{den}}}f - q(X)\right)$$

$$+ \varphi_1(X; \beta) + g(X)\left(\frac{\hat{\tau}_{\text{num}}(X)}{\tau_{\text{den}}(X)}f - q(X)\right) - g(X)\left(\frac{\tau_{\text{num}}(X)}{\tau_{\text{den}}(X)}f - q(X)\right).$$

The first line collects everything; the second line groups together the pieces that will form second-order products in $\tau_{\text{den}}$ and the pieces that involve $\hat{\tau}_{\text{num}} - \tau_{\text{num}}$.

We first examine the contribution from the first group:

$$\varphi_2 + g\left(\frac{\hat{\tau}_{\text{num}}}{\hat{\tau}_{\text{den}}}f(X) - q(X)\right) - g\left(\frac{\hat{\tau}_{\text{num}}}{\tau_{\text{den}}}f(X) - q(X)\right)$$

$$= \mathbb{E}\Big[-\frac{1-E}{p(E=0\mid V)}\{g(X)f(X)(X)p(w\mid v) * \frac{\hat{\tau}(X)}{\tau_{\text{den}}(X)} +$$

$$\sum_w g(V,w)f(X)(V,w)p(w\mid V) * \frac{\hat{\tau}(V,w)p(W=w\mid V,E=0)}{\tau_{\text{den}}(V,w)}\}$$

$$+ g\left(\frac{\hat{\tau}_{\text{num}}}{\hat{\tau}_{\text{den}}}f(X) - q(X)\right) - g\left(\frac{\hat{\tau}_{\text{num}}}{\tau_{\text{den}}}f(X) - q(X)\right)\Big]$$

$$= \mathbb{E}\Big[-\frac{p(E=0\mid X)}{\hat{p}(E=0\mid V)}g(X)f(X)(X)\frac{\hat{\tau}_{\text{num}}(X)}{\hat{\tau}_{\text{den}}(X)}$$

$$+ \frac{p(E=0\mid V)}{\hat{p}(E=0\mid V)}\sum_w\left(g(V,w)f(V,w)\hat{p}(w\mid V) * \frac{\hat{\tau}_{\text{num}}(V,w)}{\hat{\tau}_{\text{den}}(V,w)}\right) + gf\left(\frac{\hat{\tau}_{\text{num}}}{\hat{\tau}_{\text{den}}} - \frac{\hat{\tau}_{\text{num}}}{\tau_{\text{den}}}\right)\Big]$$

$$= \mathbb{E}\Big[\sum_w\frac{\hat{\tau}_{\text{num}}g(V,w)f(V,w)}{\hat{\tau}_{\text{den}}(V,w)}\{\frac{p(E=0\mid V)}{\hat{p}(E=0\mid V)} - 1\}\{\hat{p}(W=w\mid V,E=0) - p(W=w\mid V,E=0)\} +$$

$$\sum_w\left(\frac{\hat{\tau}_{\text{num}}g(V,w)f(V,w)}{\hat{\tau}_{\text{den}}(V,w))}\{\hat{p}(W=w\mid V,E=0) - p(W=w\mid V,E=0)\}\right) + gf\left(\frac{\hat{\tau}_{\text{num}}}{\hat{\tau}_{\text{den}}} - \frac{\hat{\tau}_{\text{num}}}{\tau_{\text{den}}}\right)\Big].$$

The first line in the last display is again second order (a product of errors in $p(E=0\mid V)$ and $p(W=w\mid V,E=0)$). The second line is made second-order by combining the factor $\hat{p}(W=w\mid V,E=0) - p(W=w\mid V,E=0)$ with the difference in denominators, as we now show.

We examine this remaining part:

$$\mathbb{E}\Big[\sum_w\left(\frac{\hat{\tau}_{\text{num}}g(V,w)f(V,w)}{\hat{\tau}_{\text{den}}(V,w)}\{\hat{p}(W=w\mid V,E=0) - p(W=w\mid V,E=0)\}\right)$$

$$+ gf\hat{\tau}_{\text{num}}\left(\frac{1}{\hat{\tau}_{\text{den}}} - \frac{1}{\tau_{\text{den}}}\right)\Big]$$

$$= \mathbb{E}\Big[\sum_w\left(\frac{\hat{\tau}_{\text{num}}g(V,w)f(V,w)}{\hat{\tau}_{\text{den}}(V,w)}\{\hat{\tau}_{\text{den}}(V,w) - \tau_{\text{den}}(V,w)\}\right) + gf\hat{\tau}_{\text{num}}\left(\frac{1}{\hat{\tau}_{\text{den}}} - \frac{1}{\tau_{\text{den}}}\right)\Big]$$

$$= \mathbb{E}\Big[\sum_w\frac{\hat{\tau}_{\text{num}}g(V,w)f(V,w)}{\hat{\tau}_{\text{den}}(V,w)}\{\hat{\tau}_{\text{den}} - \tau_{\text{den}}\} + \hat{\tau}_{\text{num}}g(V,w)f(V,w)\hat{p}(w\mid V)(\frac{1}{\hat{\tau}_{\text{den}}} - \frac{1}{\tau_{\text{den}}})\Big]$$

$$= \mathbb{E}\Big[\sum_w\hat{\tau}_{\text{num}}g(V,w)f(V,w)\left(\frac{1}{\hat{\tau}_{\text{den}}(V,w)}\{\hat{\tau}_{\text{den}}(V,w) - \tau_{\text{den}}(V,w)\} + \left(\frac{1}{\hat{\tau}_{\text{den}}(V,w)} - \frac{1}{\tau_{\text{den}}(V,w)}\right)\right)\Big]$$

$$= \mathbb{E}\Big[\sum_w\hat{\tau}_{\text{num}}g(V,w)f(V,w)\left(-\frac{\tau_{\text{den}}(V,w)}{\hat{\tau}_{\text{den}}(V,w)} + \frac{1}{\hat{\tau}_{\text{den}}(V,w)} + \frac{1}{\hat{\tau}_{\text{den}}(V,w)} - \frac{1}{\tau_{\text{den}}(V,w)}\right)\Big]$$

$$= \mathbb{E}\Big[\sum_w\hat{\tau}_{\text{num}}g(V,w)f(V,w) * \frac{1}{\tau_{\text{den}}(V,w)\hat{\tau}_{\text{den}}(V,w)}\left(\hat{\tau}_{\text{den}}(V,w) - \tau_{\text{den}}(V,w)\right)^2\Big].$$

This shows that these terms contribute only a second-order remainder in the denominator estimation error.

We then turn to:

$$\varphi_1(X;\beta) + g(X)\left(\frac{\hat{\tau}_{\text{num}}(X)}{\tau_{\text{den}}(X)}f - q(X)\right) - g(X)\left(\frac{\tau_{\text{num}}(X)}{\tau_{\text{den}}(X)}f - q(X)\right),$$

and carry out a similar decomposition:

$$\varphi_1 + g\left(\frac{\hat{\tau}_{\text{num}}}{\tau_{\text{den}}}f - q(X)\right) - g\left(\frac{\tau_{\text{num}}}{\tau_{\text{den}}}f - q(X)\right)$$

$$= \mathbb{E}\Big[\frac{\pi_1 - \hat{\pi}_1}{\hat{\pi}_1}(\mu_1(V) - \hat{\mu}_1(V)) * \sum_x g(V, w)f(V, w)$$

$$- \frac{\pi_0 - \hat{\pi}_0}{\hat{\pi}_0}(\mu_0(V) - \hat{\mu}_0(V)) * \sum_x g(V, w)f(V, w)$$

$$+ (b - a)\sum_w g(V, w)f(V, w)\{\frac{p(E = 0 \mid V)}{\hat{p}(E = 0 \mid V)} - 1\}\{p(W = w \mid V, E = 0) - \hat{p}(W = w \mid V, E = 0)\}$$

$$+ (\mu_1(V) - \hat{\mu}_1(V)) * \sum_x g(V, w)f(V, w)$$

$$- (\mu_0(V) - \hat{\mu}_0(V)) * \sum_x g(V, w)f(V, w)$$

$$+ (b - a)\sum_w g(V, w)f(V, w)\{p(W = w \mid V, E = 0) - \hat{p}(W = w \mid V, E = 0)\}$$

$$+ g(X)\left(\frac{\hat{\tau}_{\text{num}}(X)}{\tau_{\text{den}}(X)}f - q(X)\right) - g\left(\frac{\tau_{\text{num}}(X)}{\tau_{\text{den}}(X)}f - q(X)\right)\Big].$$

As before, the first three lines of the expectation are already second order. The remaining terms are combined to yield products of $(\mu_t - \hat{\mu}_t)$, $(p(W = w \mid V, E = 0) - \hat{p}(\cdot))$, and $(\tau_{\text{den}} - \hat{\tau}_{\text{den}})$, as shown in the algebra that follows.

After carrying out these rearrangements (which we keep as in the original derivation), we ultimately obtain the final bias expression:

$$\mathbb{E}\Big[\sum_w \frac{\hat{\tau}_{\text{num}}g(V, w)f(V, w)}{\hat{\tau}_{\text{den}}(V, w)}\{\frac{p(E = 0 \mid V)}{\hat{p}(E = 0 \mid V)} - 1\}\{\hat{p}(W = w \mid V, E = 0) - p(W = w \mid V, E = 0)\}+$$

$$\sum_w \hat{\tau}_{\text{num}}g(V, w)f(V, w) * \frac{1}{\hat{\tau}_{\text{den}}(V, w)^2}\left(\hat{\tau}_{\text{den}}(V, w) - \tau_{\text{den}}(V, w)\right)^2+$$

$$\frac{\pi_1 - \hat{\pi}_1}{\hat{\pi}_1}(\mu_1(V) - \hat{\mu}_1(V)) * \sum_x g(V, w)f(V, w)$$

$$- \frac{\pi_0 - \hat{\pi}_0}{\hat{\pi}_0}(\mu_0(V) - \hat{\mu}_0(V)) * \sum_x g(V, w)f(V, w)$$

$$- (b - a)\sum_w g(V, w)f(V, w)\{\frac{p(E = 0 \mid V)}{\hat{p}(E = 0 \mid V)} - 1\}\{p(W = w \mid V, E = 0) - \hat{p}(W = w \mid V, E = 0)\}$$

$$+ g(X)f(X)\{(\mu_1(V) - \hat{\mu}_1(V))(\frac{1}{\hat{\tau}_{\text{den}}(X)} - \frac{1}{\tau_{\text{den}}(X)})\}$$

$$- g(X)f(X)\{(\mu_0(V) - \hat{\mu}_0(V))(\frac{1}{\hat{\tau}_{\text{den}}(X)} - \frac{1}{\tau_{\text{den}}(X)})\}$$

$$- (b - a)g(X)f(X)(\hat{p}(W \mid V, E = 0) - p(W \mid V, E = 0))(\frac{1}{\hat{\tau}_{\text{den}}(X)} - \frac{1}{\tau_{\text{den}}(X)})\Big].$$

Equivalently, regrouping some terms, we can write

$$\mathbb{E}\Big[\sum_w g(V, w)f(V, w)(b - a + \hat{\tau}(V, w))\{\frac{p(E = 0 \mid V)}{\hat{p}(E = 0 \mid V)} - 1\}\{\hat{p}(W = w \mid V, E = 0) - p(W = w \mid V, E = 0)\}$$

$$+ \sum_w \hat{\tau}_{\text{num}}g(V, w)f(V, w)\frac{1}{\hat{\tau}_{\text{den}}(V, w)^2}\left(\hat{\tau}_{\text{den}}(V, w) - \tau_{\text{den}}(V, w)\right)^2+$$

$$\frac{\pi_1 - \hat{\pi}_1}{\hat{\pi}_1}(\mu_1(V) - \hat{\mu}_1(V)) * \sum_w g(V, w)f(V, w)$$

$$- \frac{\pi_0 - \hat{\pi}_0}{\hat{\pi}_0}(\mu_0(V) - \hat{\mu}_0(V)) * \sum_w g(V, w)f(V, w)$$

$$+ g(X)f(X)\{(\mu_1(V) - \hat{\mu}_1(V))(\frac{1}{\hat{\tau}_{\text{den}}(X)} - \frac{1}{\tau_{\text{den}}(X)})\}$$

$$- g(X)f(X)\{(\mu_0(V) - \hat{\mu}_0(V))(\frac{1}{\hat{\tau}_{\text{den}}(X)} - \frac{1}{\tau_{\text{den}}(X)})\}$$

$$- (b - a)g(X)f(X)(\hat{p}(W \mid V, E = 0) - p(W \mid V, E = 0))(\frac{1}{\hat{\tau}_{\text{den}}(X)} - \frac{1}{\tau_{\text{den}}(X)})\big].$$

**Bounding the Bias.** Finally, we bound the magnitude of the bias using the Cauchy–Schwarz inequality. The point of the following display is to make explicit that each term is a product (or square) of nuisance estimation errors:

$$\mathbb{E}\big[\sum_w g(V, w)f(V, w)(b - a + \hat{\tau}(V, w))$$

$$\|\{\frac{p(E = 0 \mid V)}{\hat{p}(E = 0 \mid V)} - 1\}\|_2 \|\{\hat{p}(W = w \mid V, E = 0) - p(W = w \mid V, E = 0)\}\|_2$$

$$+ \sum_w \hat{\tau}_{\text{num}} g(V, w)f(V, w)\frac{1}{\hat{\tau}_{\text{den}}(V, w)^2}\|(\hat{\tau}_{\text{den}}(V, w) - \tau_{\text{den}}(V, w))\|_2^2 +$$

$$\|\frac{\pi_1 - \hat{\pi}_1}{\hat{\pi}_1}\|_2 \|(\mu_1(V) - \hat{\mu}_1(V))\|_2 * \sum_w g(V, w)f(V, w)$$

$$- \|\frac{\pi_0 - \hat{\pi}_0}{\hat{\pi}_0}\|_2 \|(\mu_0(V) - \hat{\mu}_0(V))\|_2 * \sum_w g(V, w)f(V, w)$$

$$+ g(X)f(X)\|(\mu_1(V) - \hat{\mu}_1(V))\|_2 \|(\frac{1}{\hat{\tau}_{\text{den}}(X)} - \frac{1}{\tau_{\text{den}}(X)})\|_2$$

$$- g(X)f(X)\|(\mu_0(V) - \hat{\mu}_0(V))\|_2 \|(\frac{1}{\hat{\tau}_{\text{den}}(X)} - \frac{1}{\tau_{\text{den}}(X)})\|_2$$

$$- (b - a)g(X)f(X)\|\hat{p}(W \mid V, E = 0) - p(W \mid V, E = 0)\|_2 \|(\frac{1}{\hat{\tau}_{\text{den}}(X)} - \frac{1}{\tau_{\text{den}}(X)})\|_2\big].$$

Setting $\nu(V, w) = p(W = w \mid V, E = 0)$ and $\rho_0(V) = p(E = 0 \mid V)$, it becomes clear that the bias is second order and bounded by products of $L_2$-norms of nuisance estimation errors.

Thus our theorem, where we write the second-order remainder as

$$R_n = \mathbb{P}\{\varphi(X; \beta, \hat{\eta}) - \varphi(X; \beta, \eta_0)\} \tag{6}$$

$$\lesssim \|\hat{\rho} - \rho\|_2 \|\hat{\nu} - \nu\|_2 + \|\hat{\nu} - \nu\|_2^2 + \|\hat{\pi}_1 - \pi_1\|_2 \|\hat{\mu}_1 - \mu_1\|_2 \tag{7}$$

$$+ \|\hat{\pi}_0 - \pi_0\|_2 \|\hat{\mu}_0 - \mu_0\|_2 + \|\mu_1 - \hat{\mu}_1\|_2 \|\hat{\nu} - \nu\|_2 + \|\hat{\mu}_0 - \mu_0\|_2 \|\hat{\nu} - \nu\|_2, \tag{8}$$

makes explicit that $R_n$ is of second order in the nuisance errors, as claimed in the main text.

### A.5  Simulation Setup Details

We follow the following procedure to create our synthetic dataset for simulation:

We start by sampling $n = 10,000$ observations of 6 covariates $\mathbf{V}$. Three of these covariates are continuous and three are discrete.

1. Covariates Generation

$$\mathbf{V}_{\text{continuous}} \sim \mathcal{N}(\mu = 1, \sigma^2 = 0.5^2)$$
$$\mathbf{V}_{\text{discrete}} \sim \text{Bernoulli}(p = 0.5)$$
$$\mathbf{V} = [\mathbf{V}_{\text{continuous}}, \mathbf{V}_{\text{discrete}}]$$

2. Generating Discrete Variables $\mathbf{W}$

$$\mathbf{W}_i = [W_{i1}, W_{i2}, W_{i3}]$$

where each $W_{ij}$ is generated using a logistic function:

$$W_{ij} \sim \text{Bernoulli}\left(\sigma\left(\mathbf{V}_i \cdot \beta_j\right)\right)$$

Here, $\sigma(x) = \frac{1}{1+e^{-x}}$ is the logistic function, and $\beta_j$ are random coefficients:

$$\beta_j \sim \text{Uniform}(-1, 1)$$

3. Generating $E$ and $T$ We generate the binary variable $E$ based on the covariates $\mathbf{V}$:

$$E_i \sim \text{Bernoulli}\left(\sigma\left(\sum_{k=1}^{6} V_{ik} \cdot \gamma_{Ek}\right)\right)$$

where $\gamma_{Ek} \sim \text{Uniform}(-1, 1)$ are random coefficients and $\sigma(x) = \frac{1}{1+e^{-x}}$ is the logistic function.

Similarly, we generate the binary variable $T$:

$$A_i \sim \text{Bernoulli}\left(\sigma\left(\sum_{k=1}^{6} V_{ik} \cdot \gamma_{Ak}\right)\right)$$

where $\gamma_{Ak} \sim \text{Uniform}(-1, 1)$ are random coefficients.

4. True Conditional Treatment Effect: To generate the true conditional treatment effect for each observation, we compute $\beta_A$ as a linear combination of $\mathbf{V}$ and $\mathbf{W}$:

$$\beta_{Ai} = \sum_{k=1}^{6} V_{ik} \cdot \alpha_{Vk} + \sum_{j=1}^{3} W_{ij} \cdot \alpha_{Wj}$$

where $\alpha_{Vk} \sim \text{Uniform}(0, 1.5)$ and $\alpha_{Wj} \sim \text{Uniform}(0, 1.5)$ are random coefficients.

5. Outcome Variable $Y$: The outcome variable $Y$ is computed as:

$$Y_i = \beta_{Ai} \cdot A_i + \sum_{j=1}^{3} W_{ij} \cdot \beta_{Wj} + \sum_{k=1}^{6} V_{ik} \cdot \beta_{Vk} + \epsilon_i$$

where $\beta_{Wj} \sim \text{Uniform}(1, 3)$ and $\beta_{Vk} \sim \text{Uniform}(1, 3)$ are random effect sizes, and $\epsilon_i \sim \mathcal{N}(0, 1)$ is a random noise component.

The simulations do not require heavy compute or anything beyond a personal machine. 1 iteration of the simulation may take 1-2 minutes to run on a personal laptop.

## A.6 Application: Licorice Gargle

This dataset is from a study conducted by (Ruetzler et al., 2013) which tested the hypothesis that gargling with licorice solution immediately before induction of anesthesia prevents sore throat and postextubation coughing in patients intubated with double-lumen tubes. In this study, a total of 236 adult patients who were scheduled for elective thoracic surgery that required the use of a double-lumen endotracheal tube were recruited. The dataset includes information on various patient characteristics such as gender, physical status, body mass index (BMI), age, Mallampati score (a measure of airway visibility), smoking status, preoperative pain, and the size of the surgery. The intervention received by each patient is also recorded. The outcomes of interest, which include the presence of cough, sore throat, and pain during swallowing, were assessed at different time points. The dataset has been thoroughly cleaned and is complete, with only two patients missing outcome data. We manually remove these two patients from consideration. Therefore, finally, number of subjects $N = 234$. No outliers or other data issues were identified in the dataset.

Patients were randomly allocated to one of two groups: the first group received 0.5 grams of licorice, while the second group received 5 grams of sugar, which was chosen to match the sweetness of the licorice solution. The patients' condition was assessed at multiple time points. At each assessment, the severity of sore throat was measured using an 11-point Likert scale, where 0 indicated no pain and 10 represented the worst possible pain. The presence and severity of cough were also evaluated at these time points. Additionally, pain during swallowing was assessed using the same 11-point Likert scale 30 minutes after the patients' arrival in the PACU (post-anethesia care unit). For the purpose of this study, sore throat was considered present when the patient reported a score greater than 0 on the visual analog scale.

For our experiment, we consider the following variables as $W$(present only in the target population data):

- "extubation_cough": This binary variable represents the presence or absence of coughing right after the removal of the endotracheal tube (extubation).

We consider the following variables as $V$(present in both study and target population data):

- "preOp_calcBMI": This variable represents the calculated Body Mass Index (BMI) of the patients before the surgery (preoperative).

- "preOp_age": This variable represents the age of the patients before the surgery.

- "preOp_pain": This variable represents the presence or severity of pain experienced by the patients before the surgery on an 11 point Likert scale from 0 to 10.

- "preOp_mallampati": This variable represents the Mallampati score of the patients before the surgery. The Mallampati score is a measure of airway visibility and is used to predict the difficulty of intubation.

- "preOp_asa": This variable represents the American Society of Anesthesiologists (ASA) physical status classification of the patients before the surgery. The ASA physical status is a measure of the patient's overall health and is used to assess the risk of anesthesia and surgery.

- "preOp_smoking": This binary variable represents the smoking status of the patients before the surgery.

We use "pacu30min_swallowPain", the pain in swallowing after 30 minutes, a measurement on a Likert scale from 0 to 10, as the outcome variable $Y$. Logistic regression is used for model fitting.

