# OpenReview forum: "Accounting for Missing Covariates in Heterogeneous Treatment Estimation"
_TMLR — Accepted by TMLR_

### Review · Reviewer_nnC3 · 2025-08-25

**Summary Of Contributions:**

The paper addresses the transport of heterogeneous treatment effects when the target population contains covariates not measured in the original study. It introduces a partial-identification framework inspired by ecological inference to bound fully conditional CATEs, ensuring consistency with restricted CATEs when marginalizing over the new covariates. The authors develop a bias-corrected estimator with influence-function adjustments that achieves $\sqrt{n}$ convergence and asymptotic normality. Simulations and an RCT example demonstrate that the method yields substantially tighter and more informative bounds than baseline approaches.

**Strengths**
1. Conceptual novelty. The connection between ecological inference and causal partial identification is both creative and technically interesting.
2. Practical relevance. The setting tackled is relevant to real-world applications, such as healthcare interventions where additional covariates are often observed in deployment populations but absent in study data.
3. Empirical validation. Both simulated and real data experiments highlight the informativeness of the proposed bounds compared to natural baselines.

**Weaknesses**
1. Strong assumptions. Transport assumptions (equal distribution of W across populations and equal CATEs across domains) are restrictive and difficult to validate in practice.
2. Discrete-W requirement. The framework requires W to be discrete (or discretized), potentially limiting applicability or inducing discretization bias for inherently continuous covariates.
3. Estimator vulnerability. Performance depends critically on estimating P(W|V) in the target population; the estimator is not fully doubly robust.
4. Clarity and accessibility. Some proofs and derivations are dense, and key results could be stated more clearly in the main text for readability.
5. Minor presentation issues. Inconsistencies in notation, missing proof of Theorem 3, and figure labeling errors affect polish.

**Audience:**

Yes

**Audience Explanation:**

At least some members of the TMLR audience will be interested in this paper because it tackles a practical challenge in causal inference: transporting treatment effect estimates when new covariates appear in the target population. This setting arises often in healthcare, policy, and other applied domains where study data omit important variables. The paper’s use of ecological inference ideas for partial identification, coupled with a bias-corrected estimator, provides novel methodological contributions and practical tools for generalization and fairness—topics of broad relevance in the machine learning community.

**Broader Impact Concerns:**

No ethical concerns were identified that would require a broader impact statement.

**Claims And Evidence:**

No

**Claims Explanation:**

The authors didn't provide the proof of Theorem 3.

**Requested Changes:**

1. Discuss the strength and restrictions of the assumptions in Section 2, and compare them with assumptions used in related work.
2. Discuss whether the framework can be adapted to handle continuous W without discretization.
3. Discuss potential strategies to achieve robustness, or at least mitigate the impact, when $\hat{v}$ is misspecified.
4. The appendix is difficult to navigate; it would improve readability if theorems were stated clearly in the main text, with only the proofs deferred to the appendix and cited appropriately.
5. In Section 3.3, the presentation of the Bias-Corrected Estimator, together with Assumption (4) and the subsequent inequality, would benefit from being stated formally as a theorem in the main body. Doing so would enhance clarity, provide a clear reference point for readers, and align with the presentation style of other main theoretical results.
6. Provide the proof of Theorem 3.3.
7. Ensure consistent notation for $\phi(X, \beta, \eta)$ versus $\phi(X; \beta, \eta)$.
8. Correct the caption of Figure 3 to read Bottom (c, d) instead of Bottom (b, c).

---

> ### Author Response · Authors · 2025-11-08
> **Reviewer nnC3 response**
>
> Thank you for the suggestions! We will make all recommended changes to notation, wording and style.
>
> ## Theorem 3 Response
>
> In our text, we write that Theorem 3 follows directly from Theorem 2 together with the standard Z–estimation analysis under misspecification (cf. Lemma 3 of Kennedy et al., 2023). Mapping Lemma 3 (where we explicitly satisfy all assumptions) in Kennedy et al. (titled “Semiparametric Counterfactual Density Estimation”) to our notation ($\\theta \\equiv \\beta$, $Z \\equiv X$, $\\eta \\equiv \\eta_0$), we obtain the linear expansion
>
> $$
> \\hat\\beta-\\beta
> = -\\,M^{-1}\\,(P_n-P)\\{\\varphi(X;\\beta,\\eta_0)\\}\\;+\\;O_p(R_n)\\;+\\;o_p(n^{-1/2}).
> $$
>
> where
>
> $$
> M := \\partial_{\\beta}\\,\\mathbb{E}\\!\\left[\\varphi(X;\\beta,\\eta_0)\\right],
> \\qquad
> R_n := \\mathbb{E}\\bigl\\{\\varphi(X;\\beta,\\hat\\eta)-\\varphi(X;\\beta,\\eta_0)\\bigr\\}.
> $$
>
> Theorem 2 shows $R_n=o_p(n^{-1/2})$, so
>
> $$
> \\hat\\beta-\\beta
> = -\\,M^{-1}\\,(P_n-P)\\{\\varphi(X;\\beta,\\eta_0)\\}\\;+\\;o_p(n^{-1/2}).
> $$
>
> Multiplying by $\\sqrt{n}$ and invoking the multivariate CLT for the mean of i.i.d. mean–zero vectors $\\varphi(X;\\beta,\\eta_0)$ with finite covariance $V := \\operatorname{Var}\\!\\bigl(\\varphi(X;\\beta,\\eta_0)\\bigr)$ (equivalently, $\\|(P_n-P)\\{\\varphi\\}\\|=O_p(n^{-1/2})$ by CLT), then the continuous mapping theorem (multiplication by $-M^{-1}$) and Slutsky’s theorem yield
>
> $$
> \\sqrt{n}\\,(\\hat\\beta-\\beta) \\Rightarrow \\mathcal{N}\\!\\bigl(0,\\ M^{-1} V\\, M^{-1}\\bigr),
> $$
>
> which is exactly Theorem 3.
>
> ## Assumptions: strengths, limitations, and comparison to related work
>
> **Strengths.** The identifying content consists of: (i) study-side unconfoundedness $T \\perp (Y^0,Y^1)\\mid V$; (ii) a covariate-shift restriction $\\mathbb{P}(W\\mid V,E{=}0)=\\mathbb{P}(W\\mid V,E{=}1)$ and an effect-transport restriction $\\mathbb{E}[Y^1{-}Y^0\\mid V,W,E{=}0]=\\mathbb{E}[Y^1{-}Y^0\\mid V,W,E{=}1]$; (iii) bounded outcomes and positivity. These permit partial identification of $\\mathbb{E}[Y^1{-}Y^0\\mid V,W]$ with no outcomes in $E{=}0$, and allow flexible learning of nuisances in $E{=}1$.
>
> **Limitations.** The restriction on $\\mathbb{P}(W\\mid V,\\cdot)$ can fail if study selection acts on factors beyond $V$ that also affect $W$. The effect-transport restriction excludes residual effect modification by $E$ after conditioning on $(V,W)$.
>
> **Relation to prior work.**
> Approaches that claim point identification with missing covariates typically assume (A) measurement/proxy models that recover latent covariates via negative controls or proxy variables [1,2,3], (B) transport graphs specifying cross-domain mechanisms [4], or (C) strong overlap so all $(V,W)$ cells have outcomes [5]. Other lines combine experimental and observational outcomes on common covariates only and model confounding bias or distributional shift [6,7,8,9]. In contrast, our setting avoids these strong assumptions limits us to bounding treatment effects instead of point identification.
>
> **References**
>
> [1] Judea Pearl (2010), *On Measurement Bias in Causal Inference*.
> [2] Wang Miao, Zhi Geng, Eric Tchetgen Tchetgen (2018), *Identifying Causal Effects With Proxy Variables of an Unmeasured Confounder*.
> [3] Eric J. Tchetgen Tchetgen, Andrew Ying, Yifan Cui, Xu Shi, Wang Miao (2020), *An Introduction to Proximal Causal Learning*.
> [4] Elias Bareinboim, Judea Pearl (2014), *Transportability from Multiple Environments with Limited Experiments*.
> [5] Richard K. Crump, V. Joseph Hotz, Guido W. Imbens, Oscar A. Mitnik (2009), *Dealing with Limited Overlap in Estimation of Average Treatment Effects*.
> [6] Shuxiao Chen, Bo Zhang, Ting Ye (2021), *Minimax Rates and Adaptivity in Combining Experimental and Observational Data*.
> [7] Tobias Hatt, Jeroen Berrevoets, Alicia Curth, Stefan Feuerriegel, Mihaela van der Schaar (2022), *Combining Observational and Randomized Data for Estimating Heterogeneous Treatment Effects*.
> [8] Ilker Demirel, Ahmed Alaa, Anthony Philippakis, David Sontag (2024), *Prediction-powered Generalization of Causal Inferences*.
> [9] Lili Wu, Shu Yang (2022), *Integrative R-learner of Heterogeneous Treatment Effects Combining Experimental and Observational Studies*.
>
> ## Adaptation to Continuous $W$
>
> This is not possible under our framework. However, we believe our discretization assumption is not overly restrictive as much medical data is naturally discrete (e.g., pain scores), and furthermore any continuous data can be made discrete by bucketing.
>
> ## Mitigating misspecification of $\\nu(v,w)=\\mathbb{P}(W{=}w\\mid V{=}v,E{=}0)$
>
> Because $\\nu$ is the unique link between study and target, errors in $\\hat\\nu$ cannot be fully offset by other nuisances. One strategy practitioners could use is to try a sensitivity analysis over a range of different estimation classes. We will make the importance of $\\nu$ being properly specified clear in our paper.

---

> > ### Author Response · Authors · 2025-12-08
> > **Review Reminder**
> >
> > Hi and thanks for your hard work! We were just writing as a reminder that we had rebutted.

---

> > > ### Comment · Reviewer_nnC3 · 2025-12-09
> > > **Official response to authors**
> > >
> > > Dear authors, all of concerns have been addressed, I have recommended accepting this paper.

---

### Review · Reviewer_WQUR · 2025-09-15

**Summary Of Contributions:**

The paper examines the situation where a clinical study was used to estimate the CATE in a first step. Then in a second step, clinicians have access to a new set of covariates that was not available in the previous study. The authors derive estimators of bounds of the CATE with additional covariates, assuming that the CATE was estimated for a reduced set of covariates on a first step. They use double machine learning and sample splitting for a (potentially misspecified) semiparametric model. This allows them to derive a semi-parametric estimator with parametric convergence rate and asymptotic normality. The problem is motivated well and the solution with double machine learning appears elegant and useful.  I also very much like how it is explained what proof techniques are involved!

**Audience:**

Yes

**Audience Explanation:**

I feel like the paper addresses an important problem in practice. As such this is a very interesting paper for people in CATE estimation, such as Machine Learning researchers working in healthcare applications. In addition, the fact that general machine learning methods can be used to estimate the nuisance functions in a first step make the paper even more interesting for a general audience.

**Claims And Evidence:**

Yes

**Claims Explanation:**

The explanations in the paper are easy to follow, as the very natural "ecological" inference idea is combined with double machine learning techniques based on earlier papers. The proofs, while in need of cleaning, appear reasonable and the experiments are convincing for me.

**Requested Changes:**

I think this is an important paper and that the main part of the paper is well written and explained. However, I think it is very important that the authors improve writing in their proofs. Since the techniques are quite important, I think it would be very nice to have well-written proofs in the paper. I recommend going through the proofs as a whole carefully and improve writing, but in particular:

- Formulas seem to come out of nowhere, such as on page 15. Please preface them with text as you would do in the main part of the paper.
- Its probably not the best notation to use $\tau_n$ as the numerator of $\tau$ (since this usually means an estimator with $n$ samples).
- While throughout the paper $ \mathbb{E}[]$ is used to denote expectations, it suddenly seems that throughout the proofs $E() $ is used to denote expectations. If this is indeed true, this needs to be carefully changed, as this is very confusing with the variable E involved.
- There is an "A" that suddenly appears to come out of nowhere, is this supposed to be T? Please check this carefully.

General comments:
- Roadmap is very helpful!
- In Theorem 1, $\nu$ should be $\nu(v,w)$
- When introducing f(X), $\tau(x)$ should have capital X.
- Could you maybe add more detail on how the nuisance functions are estimated in the application?
- For the Benchmarking explanation on page 9, please use another symbol than n in "n variables from V".
- The bibliography doesn't appear consistent, sometimes with URL/ISSN/DOI, sometimes not.

If the proofs are adequately improved, I would very much like to see this paper published in TMLR.

---

> ### Author Response · Authors · 2025-11-08
> **Reviewer WQUR response**
>
> Thank you for your comments and the writing recommendations you have made! We will make the changes to the notation and proofs that you have suggested. All nuisance functions are estimated through cross-fit logistic regression (we will add further details to paper).

---

> ### Comment · Reviewer_WQUR · 2025-12-09
>
> Thanks! If my suggestions are adequately addressed I am happy with the paper.

---

> > ### Comment · Action_Editor_YjhQ · 2025-12-09
> > **Make a final recommendation**
> >
> > Dear Reviewer,
> >
> > If you are happy with the paper, could you please make a final recommendation?
> >
> > Thanks,
> > Action Editor

---

> > ### Author Response · Authors · 2025-12-09
> > **Proofs Changed**
> >
> > Hi! As per your suggestions, we have changed the proofs to add plenty of verbal commentary for the math, and we have updated the notation as per your suggestion! We have updated the pdf in the submission to reflect this. Thanks so much for the effort in reviewing

---

### Review · Reviewer_EinB · 2025-11-07

**Summary Of Contributions:**

This paper addresses the challenge in causal inference of estimating heterogeneous treatment effects (CATEs) in a target population where certain covariates are newly observed but not available in the original study population. Traditional methods fail when these new covariates are unlinked to treatment or outcome data. To bridge this gap, the authors propose a partial identification framework inspired by ecological inference, which derives sharp bounds on the fully-conditional CATE by requiring consistency with marginal distributions observed in the study. Additionally, they introduce a bias-corrected estimator based on influence functions, allowing fast convergence even with nonparametric nuisance estimators. Experimental results on both simulated and real RCT data demonstrate tighter bounds and better coverage compared to conventional approaches.

**Audience:**

Yes

**Audience Explanation:**

This paper studies the causal inference problem, which is a highly related topic to the TMLR community.

**Broader Impact Concerns:**

No ethical concerns.

**Claims And Evidence:**

Yes

**Claims Explanation:**

The claims are supported by a theoretical identification guarantee and experimental results on different datasets.

**Requested Changes:**

1.	The assumption that the distribution of new covariates W conditioned on common covariates V is the same across study and target populations seems strong and may not hold in practice, especially when there are structural differences between populations. The paper could benefit from discussing diagnostics or empirical checks for this assumption.
2.	While the method assumes W is discrete, the implications of discretizing continuous covariates (especially in high dimensions) are not thoroughly analyzed. How sensitive are the bounds to the binning scheme?
3.	The estimator lacks robustness to errors in estimating ν, which is critical to the ecological linking. Although this is acknowledged, it may limit practical deployment unless ν is modeled very accurately.
4.	The plug-in estimator used as a baseline lacks robustness and is expected to perform poorly under misestimation. Including a stronger baseline, such as doubly robust learners adapted to the restricted CATE, or recent deep learning based approaches, would provide more meaningful comparisons.
5.	(minor) The framework focuses on binary treatment and bounded continuous outcomes. Extensions to continuous treatments or multi-valued treatments are not discussed, which limits generalizability.

---

> ### Author Response · Authors · 2025-11-12
>
> Thank you for your comments and suggestions!
>
> **Diagnosing $\mathbb{P}(W\mid V,E{=}0)=\mathbb{P}(W\mid V,E{=}1)$**
>
> Because $W$ is not observed when $E{=}1$ in our setting, the conditional–independence claim $E \perp W \mid V$ is not directly testable from the available data. If there is strong reason to believe independence is in doubt, practitioners could obtain a small pilot in the study population where $W$ is measured and then test $E \perp W \mid V$ with a cross–fitted residual/orthogonal score test on that pilot set. This is analogous to standard causal analyses that rely on inherently untestable assumptions (e.g., no unobserved confounding).
>
> **Discrete-$W$ and binning sensitivity**
>
> If you split the bins of $W$ into finer bins (a refinement) and keep the same target you care about (e.g., the average effect over a fixed region $A$), then any solution you can get with the fine bins can be averaged to a valid solution with the coarse bins. That means the set of target values possible with the fine bins is contained inside the set possible with the coarse bins, so the identified interval under the finer partition is (weakly) narrower. Additionally, this goes with the caveat of being a population statement (assuming perfect knowledge of nuisances) and only compares the same target across partitions.
>
>
> **Mitigating misspecification of $\nu(v,w)=\mathbb{P}(W{=}w\mid V{=}v,E{=}0)$**
>
> Because $\nu$ is the unique link between study and target, errors in $\hat\nu$ cannot be fully offset by other nuisances. One strategy practitioners could use is to try a sensitivity analysis over a range of different estimation classes. We will make the importance of $\nu$ being properly specified clear in our paper.
>
> **Baselines**
>
> Beyond the simple plug-in, we already compare to confidence intervals for the restricted CATE (which we estimate using a DR approach) on $V$; in both simulation and the RCT, those intervals frequently fail to cover the true fully conditional effects, whereas our bounds attain near-nominal coverage at realistic sensitivity levels.
>
> **Multi Valued T**
>
> For an extension to multi-valued $T$, our paper could be extended in the following way. With $K$ arms, compare each pair $(t,t')$ by bounding $\mathbb{E}[Y^{t}-Y^{t'}\mid V,W]$ using the binary-case recipe (replace $(\mu_1-\mu_0)$ with $(\mu_t-\mu_{t'})$ and assume arm-wise positivity), then either compare all arms to a reference or declare $t$ better than $t'$ when the lower bound is $>0$. With continuous treatments, given positivity of the density $f_{T\mid V,E=1}(t\mid v)>0$, conditional treatment-effect bounds are still identified via Theorem~1 applied point-wise in $t$. For estimation, one would draw on the continuous-treatment literature to learn the study-side dose–response $m(v,t)=\mathbb{E}[Y\mid V=v,T=t,E=1]$ (e.g., generalized propensity score–based or doubly robust learners with kernel/spline smoothing in $t$), and then plug these into our ecological bound formulas with the same influence-function correction applied point-wise.

---

> > ### Author Response · Authors · 2025-12-08
> > **Rebuttal Reminder**
> >
> > Hi and thanks for your hard work! We were just writing as a reminder that we had rebutted.

---

### Decision · Action_Editor_YjhQ · 2025-12-09

**Recommendation:** Accept with minor revision

**Additional Comments:**

Reviewers ultimately converged on acceptance. Their suggestions focus on presentation: clearer notation, smoother proofs, and integrating the justification for Theorem 3 into the paper. The authors’ rebuttal addressed the substantive questions, and the remaining changes are editorial.

**Audience:**

Yes

**Audience Explanation:**

Transporting treatment effects across populations is a central problem in applied causal inference, especially in healthcare and policy settings. The paper’s use of ecological inference ideas for partial identification, combined with an efficient estimator, gives a perspective that is both timely and practically useful.

**Claims And Evidence:**

Yes

**Claims Explanation:**

The paper provides a sound partial-identification framework for transporting heterogeneous treatment effects when new covariates appear only in the target population. The main theoretical results are well supported, and the authors have clarified the remaining questions from reviewers. The influence-function estimator is justified, and both the simulations and the RCT example offer credible evidence that the proposed bounds are informative and better calibrated than baseline approaches. Reviewers agreed that the claims are supported, with only minor improvements needed in clarity and notation.